# NFYA promotes malignant behavior of triple-negative breast cancer in mice through the regulation of lipid metabolism

Nobuhiro Okada [1,2✉], Chihiro Ueki[1], Masahiro Shimazaki[3], Goki Tsujimoto[1], Susumu Kohno[4], Hayato Muranaka[4,5], Kiyotsugu Yoshikawa[6] & Chiaki Takahashi [4]

Two splicing variants exist in NFYA that exhibit high expression in many human tumour types. The balance in their expression correlates with prognosis in breast cancer, but functional differences remain unclear. Here, we demonstrate that NFYAv1, a long-form variant, upregulates the transcription of essential lipogenic enzymes ACACA and FASN to enhance the malignant behavior of triple-negative breast cancer (TNBC). Loss of the NFYAv1-lipogenesis axis strongly suppresses malignant behavior in vitro and in vivo, indicating that the NFYAv1-lipogenesis axis is essential for TNBC malignant behavior and that the axis might be a potential therapeutic target for TNBC. Furthermore, mice deficient in lipogenic enzymes, such as Acly, Acaca, and Fasn, exhibit embryonic lethality; however, Nfyav1-deficient mice exhibited no apparent developmental abnormalities. Our results indicate that the NFYAv1-lipogenesis axis has tumour-promoting effects and that NFYAv1 may be a safe therapeutic target for TNBC.

[1] Graduate School of Interdisciplinary Science & Engineering in Health Systems, Okayama University, Okayama 700-8530, Japan. [2] Department of Pharmacology, Kyoto Prefectural University of Medicine, Kyoto 602-8566, Japan. [3] Laboratory for Malignancy Control Research, Medical Innovation Center, Kyoto University, Kyoto 606-8501, Japan. [4] Division of Oncology and Molecular Biology, Cancer Research Institute, Kanazawa University, Kanazawa 920-1192, Japan. [5] Samuel Oschin Cancer Center, Cedars-Sinai Medical Center, Los Angeles, CA 90048, USA. [6] Faculty of Pharmaceutical Sciences, Doshisha Women's College of Liberal Arts, Kyoto 610-0395, Japan. ✉email: okadan@koto.kpu-m.ac.jp

Triple-negative breast cancer (TNBC) is an aggressive breast cancer subtype that lacks the expression of the estrogen receptor, progesterone receptor, and HER2 protein[1]. TNBC is sensitive to chemotherapy but not endocrine therapy and has no identified molecular targets, resulting in a worse patient prognosis. TNBCs are classified into four subtypes according to their genetic profile: basal-like immunosuppressed, basal-like immune-activated, mesenchymal, and luminal androgen receptor[2]. In particular, the mesenchymal TNBC subtype is associated with a worse patient prognosis[3–6]. Therefore, there is an urgent need to develop new treatment regimens or to identify molecular targets for therapy. Recent advances in cancer metabolism research have revealed that reprogramming lipid metabolism is one of the hallmarks of cancer[7,8]. Lipogenic enzymes are highly expressed in TNBC to obtain membrane phospholipids, signaling molecules, protein modifications, and energy for high proliferation[7,9–12]. Therefore, preclinical and clinical trials targeting lipid metabolism as a therapeutic target are proceeding[8,13,14]. However, the selection of patients who will benefit from lipid metabolism blockade, drug resistance, and other issues remains unresolved. Furthermore, a critical challenge is that lipogenic enzymes, Acly, Acaca, or Fasn, knockout mice have been reported to show embryonic lethality in early embryogenesis, raising safety concerns regarding blocking of the lipogenesis pathway by inhibiting these genes as a therapeutic target[15–17]. Therefore, identifying a safer target that is expected to have therapeutic effects comparable to the inhibition of these enzymes is important.

NF-Y is a trimeric protein complex comprising three subunits, NFYA, NFYB, and NFYC, which bind to the CCAAT box on the promoters of target genes to activate transcription[18,19]. NFYA is considered a limiting subunit of the trimeric complex because it has a DNA-binding domain at the C-terminus, and point mutations in the DNA-binding domain inhibit NF-Y-DNA complex formation[20]. There are two types of splice variants of NFYA: the long-form (NFYAv1) and short-form (NFYAv2), which lack the 29 amino acids encoded by exon 3 in the Q-rich transactivation domain[21]. Genome-wide studies have shown that CCAAT boxes and NF-Y binding are enriched at the promoters of genes overexpressed in various types of cancer, thereby eliciting various biological effects including metabolic regulation[18,22]. Clinical studies have shown that patients with overexpression of NF-Y target genes in several types of cancers have a poor prognosis[18,23,24]. Integrative analysis of NF-Y genome binding and gene expression profiles by knockdown of NF-Y subunits revealed that NF-Y regulates de novo biosynthetic pathways of lipids[22]. Genome-wide scans showed that NF-Y overlaps with 32% of sterol regulatory element-binding protein1 (SREBP1) targets, suggesting that the effects of NF-Y in regulating lipid metabolism are coordinated with SREBP1[25]. NF-Y also activates glycolytic genes, represses mitochondrial respiratory genes, and targets genes involved in the serine, one-carbon, glycine pathway, glutamine pathway, and polyamine and purine biosynthesis, indicating that metabolic pathways are globally altered and specific cancer-causing nodes are under NF-Y control[22,26]. High levels of NFYA expression have been detected in various human cancers, including breast, prostate, gastric adenocarcinoma, and lung squamous cell carcinomas[27–30]. The expression balance of the two splice variants of NFYA varies in a tissue- and cell-specific manner, and a correlation between the expression balance and prognosis has been reported in breast cancer[21,28]. Thus, NF-Y is involved in oncogenic mechanisms and metabolic pathway regulation; however, these studies are based primarily on systematic investigations of global gene expression in large cohorts of cancer patients and cancer cells. The detailed function of NFYA in carcinogenesis and malignant progression, based on genetic and biochemical methods, remains unclear.

The current study investigated the functional importance of NFYA splicing variants in TNBC. The role of NFYAv1 in the malignant behavior of TNBC, in particular, the transcription of lipogenic enzymes ACACA and FASN, was evaluated using in vitro and in vivo models. Furthermore, tumour suppression via suppression of ACACA and FASN in breast cancer tissues and potential developmental abnormalities were investigated in Nfyav1-specific knockout mice. Our studies provide insights into the tumour-promoting effects of the NFYAv1-lipogenesis axis and indicate that NFYAv1 may be helpful as a therapeutic marker and target for TNBC.

## Results

### NFYA switches the expression of alternative splicing variants during EMT progression.

Emerging evidence suggests that splice variants of NFYA are essential components of oncogenic signals in the development of breast cancers[28,31]. In the present study, we also found that compared with non-transformed mouse mammary epithelial cells (NMuMG cells), breast cancer cells isolated from mouse breast tumour tissue showed significantly increased expression of both NFYA variants (Fig. 1b, c). Although these results suggest that NFYA plays an essential role in the malignant behavior of breast cancer, the functional differences between NFYA variants remain unclear because of the absence of functional domains in the spliced exon (Fig. 1a). To investigate the functional differences between variants, we first examined the expression patterns of the variants in various subtypes of breast cancer cells. Luminal and HER2-positive breast cancer cells predominantly expressed NFYAv2 but had almost undetectable NFYAv1 expression, which was almost the same in normal and basal-like breast cancer cells; NFYAv1 expression was slightly detectable. In contrast, Claudin$^{low}$ breast cancer cells predominantly expressed NFYAv1 but had almost undetectable NFYAv2 expression (Fig. 1d, e). The expression pattern of NFYA is associated with epithelial-mesenchymal transition (EMT), in which NFYAv2 showed the same expression pattern as an epithelial marker, E-cadherin/CDH1. NFYAv1 showed the same expression pattern as a mesenchymal marker, vimentin (Fig. 1e and Supplementary Fig. 1a). We then used normal human mammary epithelial cells (HMLE cells) and epithelial breast cancer cells (MDA-MB-468 cells) to confirm that conversion of the NFYA splicing variant expression is not only dependent on EMT status between cell lines, but also occurs with intracellular EMT progression. We sorted CD44$^{low}$/CD24$^{high}$ (HMLE-Epi cells) and CD44$^{high}$/CD24$^{low}$ (HMLE-Mes cells) cells from HMLE cells and examined the EMT status and NFYA expression. Similar to the results of breast cancer cell lines, NFYAv1 expression correlated with vimentin, whereas NFYAv2 correlated with E-cadherin (Supplementary Fig. 1b, c). Moreover, we observed that NFYA expression shifted from NFYAv2 to NFYAv1 with EMT progression induced by overexpressing SNAIL in MDA-MB-468 and HMLE cells (Fig. 1f and Supplementary Fig. 1d, e). Next, we examined whether the shift in NFYA expression is a reversible reaction. SUM159 cells were treated with forskolin, which is known to induce mesenchymal–epithelial transition (MET)[32]. We found that forskolin treatment decreased the expression of the mesenchymal marker vimentin, whereas the expression of the epithelial marker E-cadherin remained undetectable. These results indicate that forskolin treatment induces MET; however, its partial effect does not lead to complete MET induction. Despite partial MET induction, the ratio of NFYAv2 to NFYAv1 increased, indicating that the switching of NFYA variant expression is reversible (Supplementary Fig. 1f). In addition, NFYA deficiency in HMLE cells did not affect SNAIL-induced EMT progression, confirming that NFYA is not an EMT regulator but a factor that functions by

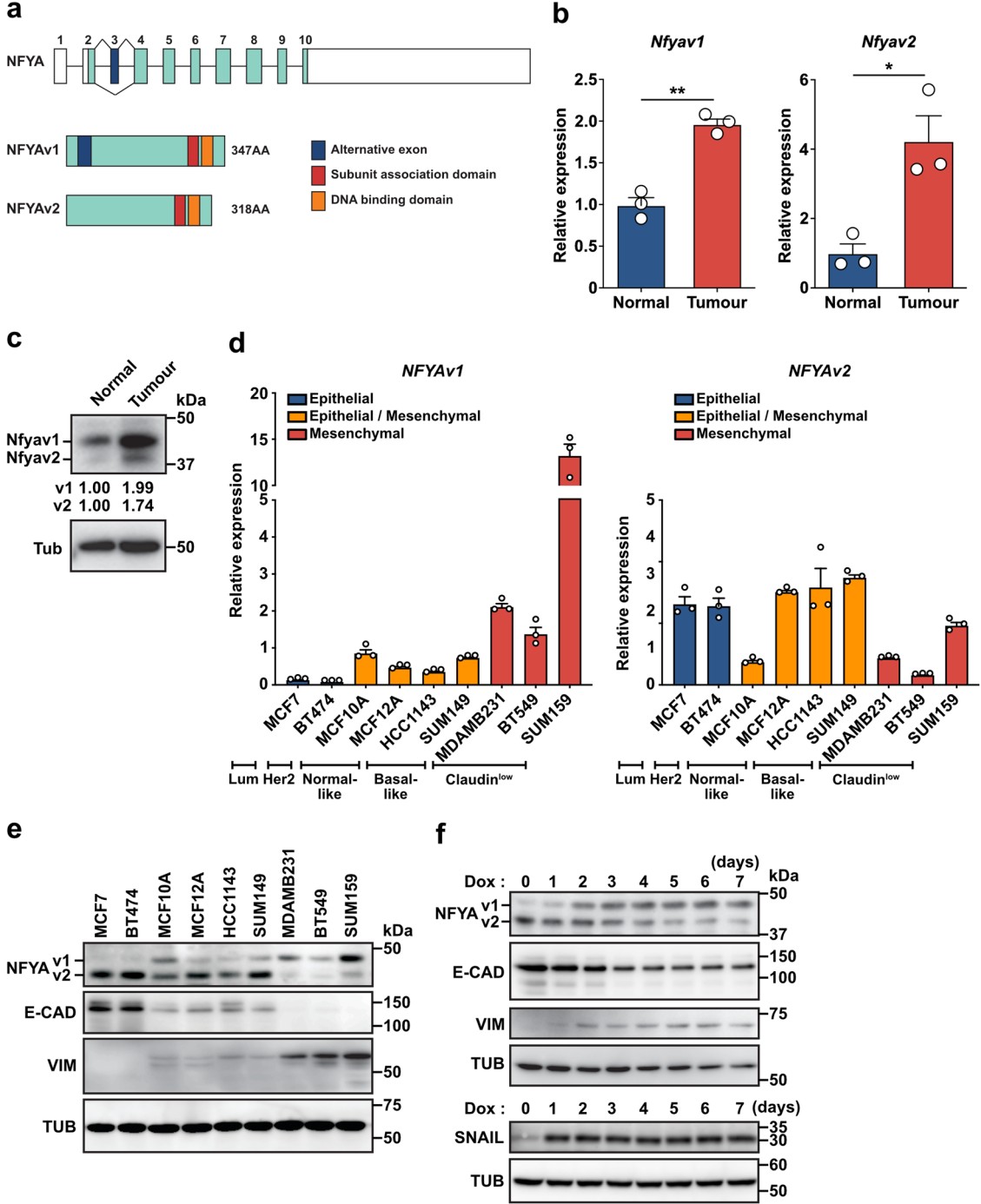

**Fig. 1 NFYA switches the expression of alternative splicing variants during EMT progression. a** Schematic diagram of the intron-exon structure of the *NFYA* gene. Alternative splicing of the third exon generates two different variants, long-form (v1) and short-form (v2). **b, c** qRT-PCR (**b**) and western blot analysis (**c**) of Nfyav1 and Nfyav2 in breast cancer cells (primary cells isolated from mouse breast cancer tissue) compared with non-transformed mouse mammary epithelial cells (NMuMG cells). $N = 3$ biologically independent experiments for qRT-PCR. **d** qRT-PCR analysis of *NFYAv1* (left panel) and *NFYAv2* (right panel) mRNA levels in various breast cancer cell lines. $N = 3$ biologically independent experiments. **e** Western blot analysis of NFYA, epithelial marker (E-CAD), and mesenchymal marker (VIM) protein levels in various breast cancer cell lines. **f** MDA-MB-468 cells were infected with lentivirus to express tet-on SNAIL and induced SNAIL expression by 1 µg/ml of doxycycline (Dox). Western blot analysis of NFYA protein level in the cells from day 0 to day 7 post-induction. E-CAD and VIM are markers for epithelial and mesenchymal cells, respectively. All error bars represent SEM; *$P < 0.05$; **$P < 0.01$.

shifting EMT-dependent splicing variant expression (Supplementary Fig. 1g).

**NFYAv1 deficiency inhibits tumour cell growth and tumorigenesis in TNBC.** TNBCs account for 10–15% of all breast

cancers. Because treatment of TNBC is currently limited to chemotherapy, TNBC has a worse prognosis than other breast cancers[1,33,34]. Therefore, to evaluate the potential of the NFYA-related pathway as a novel therapeutic target for TNBC, we focused on the functional analysis of NFYA in TNBC, namely NFYAv1. Given the increased expression of NFYA in breast

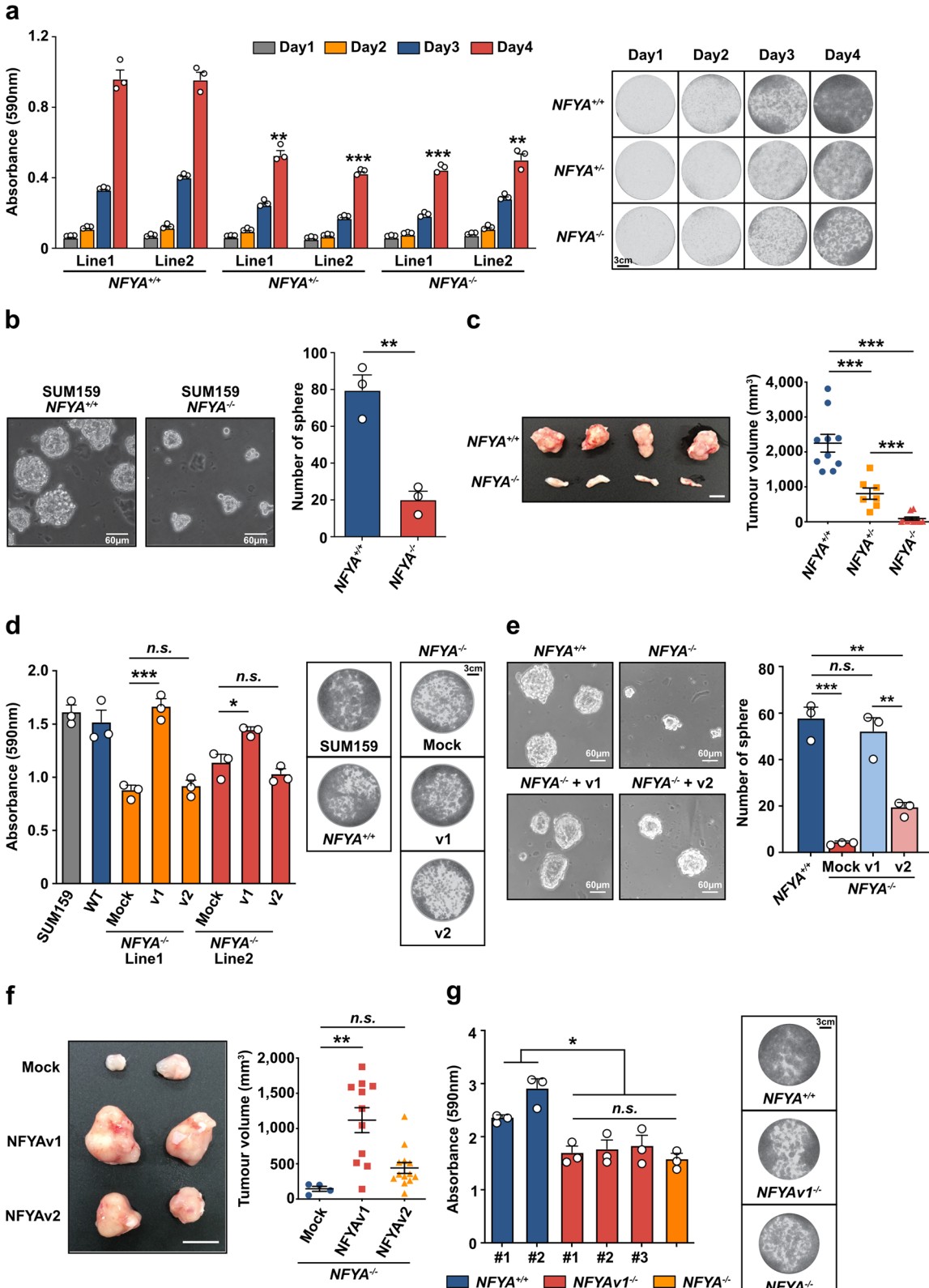

cancer cells (Fig. 1b, c), we first generated NFYA-deficient TNBC cells using the CRISPR/Cas9 system in SUM159 cells (Supplementary Fig. 2a–d) and examined whether NFYA contributes to the malignant behavior of TNBC. NFYA-deficient cells showed a repressed carcinogenic phenotype, represented by a suppressed ability to grow cells and form spheres and tumours (Fig. 2a–c). Re-expression of NFYAv1 in NFYA-deficient cells restored the

malignant behavior. However, re-expression of NFYAv2 in NFYA-deficient cells resulted in slight or no restoration of malignant behavior (Fig. 2d–f and Supplementary Fig. 3a). To further investigate the NFYA variant requirements for the malignant behavior of TNBC, we also generated NFYAv1-specific deficient SUM159 cells or overexpressed dominant-negative NFYA mutants in SUM159 cells. Cell growth was suppressed in

**Fig. 2 NFYAv1 deficiency inhibits tumour cell growth and tumorigenesis in TNBC. a** Quantification and representative pictures of 0.5 % crystal violet staining of $NFYA^{+/+}$, $NFYA^{+/-}$, and $NFYA^{-/-}$ SUM159 cells between days 1 and 4. $N = 3$ biologically independent experiments. Scale bar indicates 3 cm. **b** Representative images of sphere formation by $NFYA^{+/+}$ and $NFYA^{-/-}$ SUM159 cells. A bar graph shows the number of spheres larger than 60 μm in each group. $N = 3$ biologically independent experiments. Scale bars indicate 60 μm. **c** Representative image of tumours formed in mammary fat pads of NOD/SCID mice 50 days after implanted with $NFYA^{+/+}$ and $NFYA^{-/-}$ SUM159 cells. Dot plots show the volume of tumours from each experimental group. $N = 10$ for $NFYA^{+/+}$ and $NFYA^{-/-}$, $N = 7$ for $NFYA^{+/-}$ biologically independent experiments. Scale bar indicates 10 mm. **d** Representative images of 0.5 % crystal violet staining of $NFYA^{-/-}$ SUM159 cells overexpressed each variant of NFYA on day 4. A bar graph shows the quantification of the staining. $N = 3$ biologically independent experiments. Scale bar indicates 3 cm. **e** Representative images of sphere formation by $NFYA^{-/-}$ SUM159 cells overexpressed each variant of NFYA. A bar graph shows the number of spheres larger than 60 μm in each group. $N = 3$ biologically independent experiments. Scale bars indicate 60 μm. **f** Representative image of tumours formed in mammary fat pads of NOD/SCID mice 50 days after implanted with $NFYA^{-/-}$ SUM159 cells overexpressed each variant of NFYA. Dot plots show the volume of tumours from each experimental group. $N = 4$ for Mock, $N = 11$ for NFYAv1, $N = 13$ for NFYAv2 biologically independent experiments. Scale bar indicates 10 mm. **g** Representative images of 0.5 % crystal violet staining of two lines of $NFYA^{+/+}$, three $NFYAv1^{-/-}$, and $NFYA^{-/-}$ SUM159 cells on day 4. A bar graph shows the quantification of the staining. $N = 3$ biologically independent experiments. Scale bar indicates 3 cm. All error bars represent SEM; n.s. not significant; $*P < 0.05$; $**P < 0.01$; $***P < 0.001$.

NFYAv1-specific deficient cells, similar to that in NFYA-deficient cells (Fig. 2g), even though they expressed higher levels of NFYAv2 than control cells (Supplementary Fig. 3b). Mutations within the DNA-binding domain of NFYA have been shown to act as a dominant repressor of NF-Y-DNA complex formation and NF-Y-dependent transcription[20]. Overexpression of the dominant-negative mutant of NFYAv1 significantly suppressed cell growth, whereas the effect of the dominant-negative mutant of NFYAv2 was only partial (Supplementary Fig. 3c). These observations suggest that NFYAv1, but not NFYAv2, is functionally essential for the malignant behavior of TNBC. In addition, functional differences among NFYA splicing variants might not be limited to TNBC but also general in breast cancers because NFYA deficiency does not affect the cell growth and sphere formation in HMLE and MCF7 cells predominantly expressing NFYAv2 (Supplementary Figs. 1g and 3d–f).

**NFYA regulates lipid metabolism for the malignant behavior of TNBC.** Some studies that analyzed global gene expression in cancer cell lines containing colon, liver, and lung cancers have suggested that the NF-Y complex controls de novo lipogenesis with SREBPs[22,25]. Furthermore, we showed that the expression of lipogenic enzymes, Acly, Acaca, and Fasn, increased in breast cancer cells, and inhibition of lipogenesis by the FASN inhibitor cerulenin completely inhibited TNBC cell growth and the sphere-forming ability (Supplementary Fig. 4a–c). These findings led us to hypothesize that NFYA plays an essential role in the malignant behavior of TNBC through the regulation of lipogenesis. To test this hypothesis, we first examined whether the suppression of malignant behavior induced by NFYA deficiency could be attributed to lipid deficiency. Addition of lipids to the culture medium restored cell proliferation and sphere-forming ability of NFYA-deficient cells (Fig. 3a, b). To confirm that the restoration of malignant behavior by lipid addition is not a cell line-specific effect but a general effect on TNBCs, we generated another NFYA-deficient TNBC cell line using BT549 cells and confirmed their malignant behavior. The results showed that NFYA-deficient BT549 cells exhibited phenotypes identical to those of NFYA-deficient SUM159 cells, indicating that these phenotypes are not cell line-specific but are general to TNBCs (Supplementary Fig. 5a, b). Moreover, NFYA deficiency markedly reduced the number of lipid droplets, and addition of lipids restored the number of lipid droplets (Fig. 3c). These results suggest that NFYA positively regulates lipogenesis in the malignant behavior of TNBC and that the loss of NFYA disrupts this mechanism. We also investigated these phenotypic properties by measuring the oxygen consumption rate (OCR) since de novo fatty acid synthesis is traded off with fatty acid oxidation (FAO) (Fig. 3d, e). It is because malonyl-CoA inhibits carnitine palmitoyl transferase

(CPT), which is the rate-limiting enzyme of FAO. Long-chain fatty acids, the major substrates fueling TCA cycle via degradation by FAO, increase the OCR. Therefore, we evaluated the fatty acid consumption by measuring the increase in OCR with the addition of BSA-conjugated palmitate. Under normal conditions, wild-type NFYA cells respond to palmitate supplementation by incrementing OCR, while NFYA-deficient cells showed less response to supplementation. These results indicated that FAO was limited by fatty acid synthesis and that NFYA-deficient cells used glucose as the carbon source in the TCA cycle under normal conditions. A more significant difference was observed in maximal respiration under high-energy demand conditions with uncoupling agent FCCP treatment. Under maximal respiration conditions, reserved respiration capacity was not changed between NFYA wild-type and deficient cells, indicating that FAO activity is retained in NFYA-deficient cells. Although our results suggested that NFYA promotes malignant behavior of TNBC by regulating lipogenesis, the possibility that NFYA deficiency disrupts glucose metabolism and the TCA cycle, which supplies acetyl-CoA, the source of lipogenesis, needs to be eliminated (Supplementary Fig. 5c)[35,36]. We therefore evaluated cell behavior by adding exogenous sodium acetate to supply acetyl-CoA without mitochondrial metabolism. The addition of sodium acetate had no effects on cell growth or lipid droplet accumulation (Supplementary Fig. 5d, e). Finally, we evaluated cell growth following treatment with a CPT1A inhibitor etomoxir to corroborate that the restoration of malignant behavior in NFYA-deficient cells by adding lipids is indeed a response mediated by FAO. The results showed that CPT1A inhibition completely abolished the effects of lipid addition (Fig. 3f, g).

**NFYA enhances lipogenesis by transcriptional activation of ACACA and FASN.** Given that NFYA promotes the malignant behavior of TNBC by accelerating lipid synthesis, we next examined the expression of lipid metabolism-related genes to determine how NFYA regulates lipid metabolism, consisting of de novo lipogenesis and FAO, involving multiple enzymatic reactions (Fig. 4a). In NFYA-deficient cells, the expression of lipogenesis-related genes, including ACLY, ACACA, ACACB, and FASN, and FAO-related genes, including CPT1A and ACADL, significantly decreased (Fig. 4b, c). Furthermore, addition of lipids increased the expression of CPT1A and ACADL, even in NFYA-deficient cells (Fig. 4b, c). Because the expression of ACLY and ACACB also increased in NFYA-deficient cells with the addition of lipids, we excluded them from this study as potential NFYA targets at this time. We confirmed the protein levels of ACACA and FASN using western blotting and immunofluorescence staining. Similar results were obtained for mRNA expression (Fig. 4d and Supplementary Fig. 6a–c). Similar to the

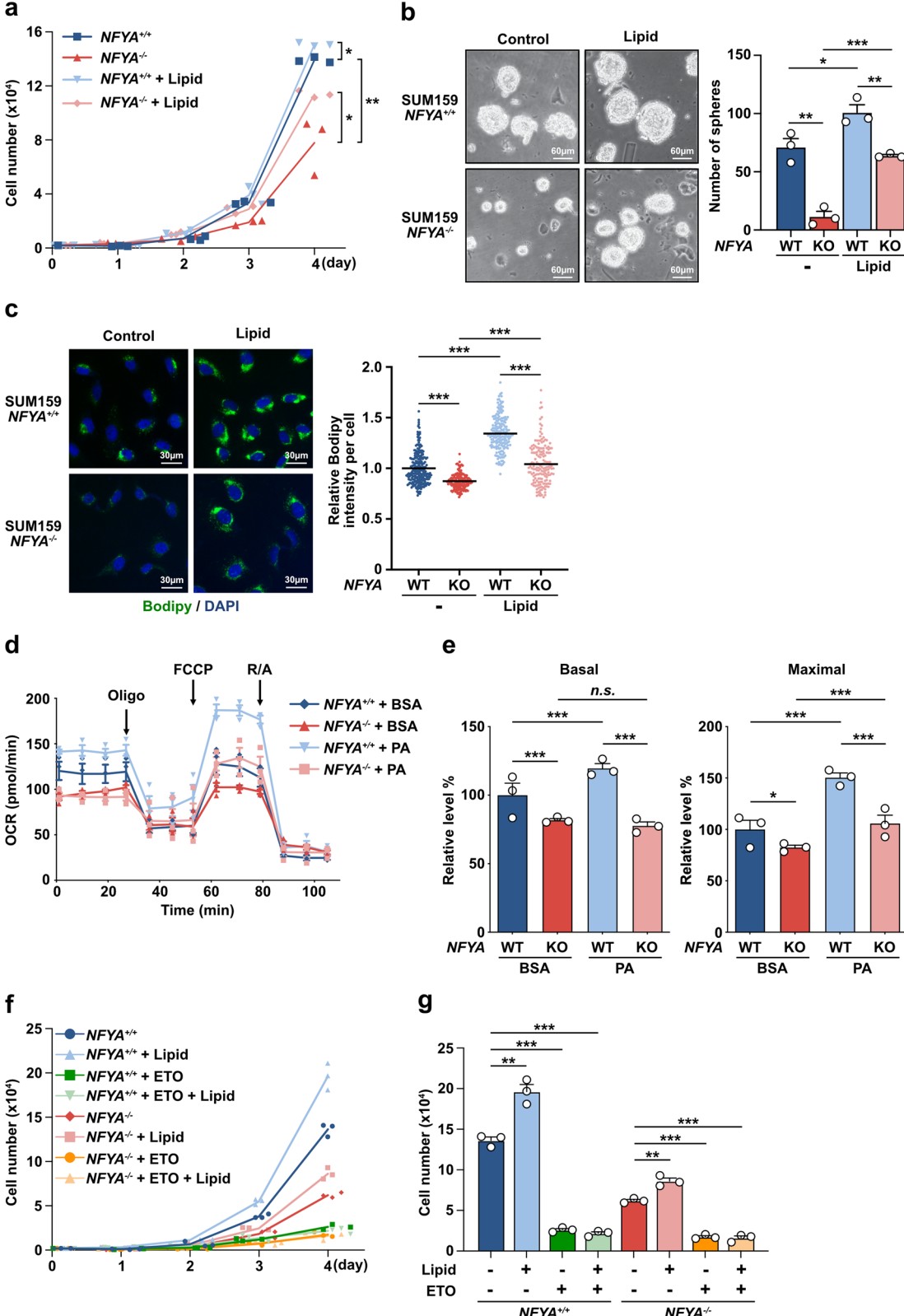

results for the cell phenotype (Supplementary Fig. 5d, e), the addition of sodium acetate had no effect on either wild-type or NFYA-deficient cells (Supplementary Fig. 6d). As the inhibition of CPT1A by etomoxir completely abolished the effects of lipid addition on proliferation enhancement (Fig. 3f, g), we analyzed the expression of FAO-related factors to confirm that FAO was suppressed by etomoxir treatment. In both wild-type and NFYA-

deficient cells, etomoxir treatment unexpectedly increased CPT1A expression (Supplementary Fig. 6e). However, no difference in the expression pattern was observed with etomoxir treatment in the presence or absence of NFYA. It has been reported that increased CPT1A expression with etomoxir treatment may reflect a compensatory response to FAO suppression, but it is inactive[37–39]. In contrast, the FASN inhibitor cerulenin

**Fig. 3 NFYA regulates lipid metabolism for malignant behavior of TNBC. a** Cumulative population of cells was measured for 4 consecutive days in $NFYA^{+/+}$ and $NFYA^{-/-}$ SUM159 cells with or without the addition of lipid mixture. $N = 3$ biologically independent experiments. **b** Representative images of sphere formation by $NFYA^{+/+}$ and $NFYA^{-/-}$ SUM159 cells with or without adding a lipid mixture. A bar graph shows the number of spheres larger than 60 μm in each group. $N = 3$ biologically independent experiments. Scale bars indicate 60 μm. **c** Representative fluorescence images of lipid droplet (green) detected with Bodipy 493/503 and nucleus (blue) detected with DAPI in $NFYA^{+/+}$ and $NFYA^{-/-}$ SUM159 cells with or without the addition of lipid mixture for 2 days. The dot plot shows the bodipy intensity per cell. $N = 247$ for $NFYA^{+/+}$, $N = 157$ for $NFYA^{-/-}$, $N = 192$ for $NFYA^{+/+}$ with lipid, $N = 166$ for $NFYA^{-/-}$ with lipid, from three biologically independent experiments. Scale bars indicate 30 μm. Means represent SEM. **d** Oxygen consumption rate (OCR) (mean ± SEM) in $NFYA^{+/+}$ and $NFYA^{-/-}$ SUM159 cells treated with the long-chain fatty acid palmitate (PA) or BSA. $N = 3$ biologically independent experiments. Oligo; oligomycin, FCCP; carbonyl cyanide-4-trifluoro methoxy phenyl hydrazone, R/A; Rotenone/antimycin A. **e** Experiments shown in Fig. 3d were quantified, and the relative levels of OCR associated with basal FAO and maximal FAO were calculated. **f** Cumulative population of cells was measured for 4 consecutive days in $NFYA^{+/+}$ and $NFYA^{-/-}$ SUM159 cells treated with or without lipid mixture, Etomoxir (ETO), and both. $N = 3$ biologically independent experiments. **g** A bar graph shows the cell number on day 4 shown in Fig. 3f. All error bars represent SEM; n.s. not significant; *$P < 0.05$; **$P < 0.01$; ***$P < 0.001$.

reduced CPT1A expression only in NFYA wild-type cells, suggesting that NFYA regulates breast cancer malignant behavior by controlling de novo lipogenesis by regulating ACACA and FASN expression.

We further examined whether NFYA regulates transcription of ACACA and FASN through direct binding to these promoter regions by CUT&RUN assay using an immunoprecipitation-validated antibody (Supplementary Fig. 6f). As NFYA binds to the CCAAT box in the promoter regions of target genes[19], we searched for the CCAAT box in the promoter regions. We found ACACA 2,825 bp upstream and FASN 577 bp and 581 bp upstream (Supplementary Fig. 6g). Analysis of NFYA binding to this region showed that NFYA binds directly to the promoter regions of ACACA and FASN (Fig. 4e). Addition of lipids to the culture medium completely abolished the binding of NFYA to the promoter regions. Furthermore, overexpression of NFYAv1, but not NFYAv2, in NFYA-deficient cells upregulated FASN, ACACA, and CPT1A expression, supporting the evidence that NFYAv1 regulates lipid metabolism via transcriptional regulation of FASN and ACACA (Fig. 4f). These results indicate that NFYA regulates the expression of ACACA and FASN by direct binding to their promoter regions, and that the lipid synthesis is suppressed in the presence of excess lipids via loss of NFYA binding. We further found that NFYA deficiency and the addition of lipids increased the expression of fatty acid translocase CD36 (Supplementary Fig. 7a, b). These results suggest that the amount of lipid in the vicinity of TNBC cells tightly regulates the balance between lipid synthesis and uptake.

**NFYA cooperatively regulates the expression of ACACA and FASN with SREBP1.** Although systematic global gene expression analysis has shown that NF-Y and SREBP1 cooperatively regulate de novo lipogenesis, the interaction between NF-Y and SREBP1 remains unclear[25]. To elucidate this interaction, we evaluated SREBP1 expression in NFYA-deficient cells. We observed increased transcription of *SREBF1* encoding SREBP1 in NFYA-deficient cells compared to that in NFYA wild-type cells, and a marked enhancement of the nuclear form of SREBP1 (Supplementary Fig. 8a, c). Furthermore, analysis of lipogenic enzyme expression under these conditions revealed decreased expression of ACACA and FASN, despite the increased activity of SREBP1 (Supplementary Fig. 8b, c). Treatment with SREBP1 inhibitor betulin also decreased the expression of these enzymes in the presence of NFYA (Supplementary Fig. 8b, c). These results indicate that NFYA and SREBP1 cooperatively regulate the expression of these enzymes and that dysfunction of either NFYA or SREBP1 results in deregulation of the expression of these enzymes.

**Nfyav1 enhances tumorigenesis via the regulation of ACACA and Fasn expression in vivo.** To investigate whether the regulation of lipid metabolism by NFYAv1 is involved in breast cancer tumorigenesis in vivo, we generated Nfyav1-specific knockout mice using the CRISPR/Cas9 system (Fig. 5a). We designed gRNAs on introns 2 and 3, introduced them into zygotes simultaneously with Cas9 protein by electroporation, and transplanted the treated embryos into pseudopregnant recipient mice to obtain founder (F0) mice. To screen Nfyav1-deficient mice, we analyzed F0 mice using genomic DNA genotyping and sequencing analysis (Supplementary Fig. 9a, b). Total knockout of Nfya has been reported to cause embryonic lethality in early development[40], and knockout of the lipogenic enzymes Acly, Acaca, and Fasn also causes embryonic lethality in early development[15–17]. Therefore, Nfyav1 knockout mice were also expected to show embryonic lethality; however, Nfyav1 knockout mice were born normal (Supplementary Table 1). We crossed F0 mice deficient in Nfyav1 with wild-type B6 mice and used the mice obtained after the F1 generation for further analysis. The loss of Nfyav1 mRNA and protein expression in $Nfyav1^{-/-}$ mice was confirmed by qRT-PCR and western blotting of mammary epithelial cells and mouse embryonic fibroblasts (Fig. 5b, c and Supplementary Fig. 9c, d). $Nfyav1^{-/-}$ mice showed a deletion of Nfyav1 expression and a significant increase in Nfyav2 expression. Although this expression was similar to that in NFYAv1-deficient cells, which suppress TNBC cell proliferation (Fig. 2g and Supplementary Fig. 3b), $Nfyav1^{-/-}$ mice were born at the expected Mendelian ratio, with no differences between the sexes (Fig. 5d). In addition, we explored the physical characteristics of $Nfyav1^{-/-}$ mice and found no apparent abnormalities in weight, mammary gland morphology, or survival (Fig. 5e, f and Supplementary Fig. 9e). These results suggest that NFYAv1 inhibition might be a safer therapeutic target than blocking lipogenesis by inhibiting lipogenic enzymes.

To investigate the effects of Nfyav1 on mammary gland tumorigenesis, we generated a genetically engineered mouse model of breast cancer based on the widely employed MMTV-PyMT mice[41,42] intercrossed with $Nfyav1^{-/-}$ mice. In MMTV-PyMT; $Nfyav1^{+/+}$ mice, mammary tumours developed as early as 41 days and 50% of the mice developed tumours by 74 days. In contrast, in MMTV-PyMT; $Nfyav1^{-/-}$ mice, the first tumour was identified at 71 days, and 50% of the mice developed tumours by 91 days (Fig. 5g). Consistent with the delay in tumour occurrence, whole-mount carmine alum staining at 15 weeks showed that MMTV-PyMT; $Nfyav1^{-/-}$ mice possessed a tumour area reduced to 29.6% of that of MMTV-PyMT; $Nfyav1^{+/+}$ mice (Fig. 5h and Supplementary Fig. 9f). Immunohistochemistry with Acaca and Fasn antibodies in tumour tissues showed high expression of these genes in MMTV-PyMT; $Nfyav1^{+/+}$ mice. In contrast, the expression of Acaca and Fasn markedly decreased in tumour tissues from MMTV-PyMT; $Nfyav1^{-/-}$ mice (Fig. 5i).

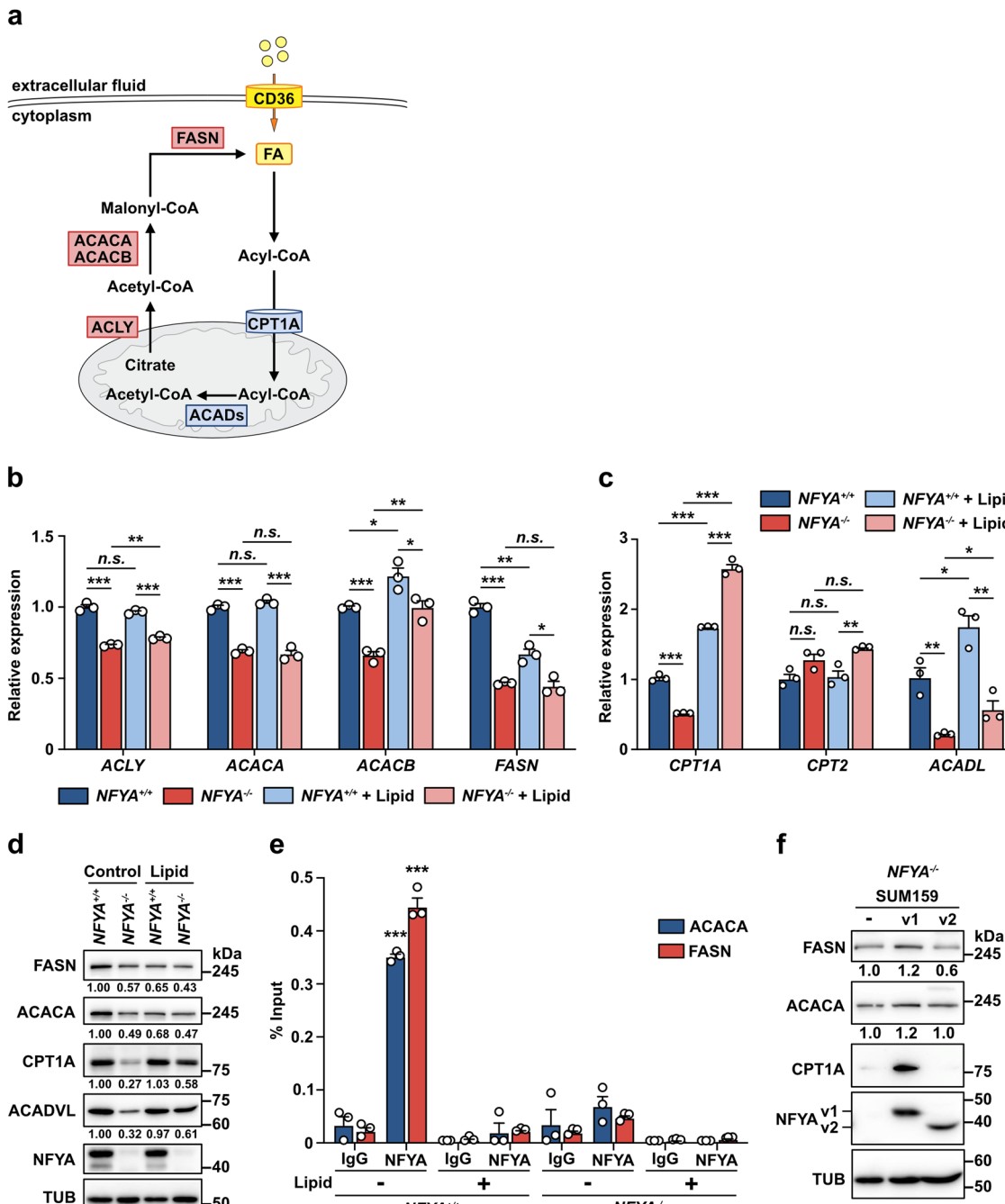

**Fig. 4 NFYA enhances lipogenesis by transcriptional activation of ACACA and FASN. a** A diagram illustrating the reaction of lipogenesis and FAO.
**b–d** qRT-PCR (**b, c**) and western blot analysis (**d**) of the expression levels of differentially expressed genes (**b**; Lipogenesis-related genes, **c**; FAO-related genes) in $NFYA^{+/+}$ and $NFYA^{-/-}$ SUM159 cells treated with or without lipid mixture. $N = 3$ biologically independent experiments for qRT-PCR.
**e** CUT&RUN assay to evaluate the association of NFYA with the promoter of ACACA and FASN in $NFYA^{+/+}$ and $NFYA^{-/-}$ SUM159 cells treated with or without lipid mixture. Input or eluted chromatin was subjected to qRT-PCR analysis using promoter-specific primers. Data represents the % input of the immunoprecipitated chromatin for each gene. $N = 3$ biologically independent experiments. **f** Western blot analysis of FASN, ACACA, and CPT1A expression in NFYAv1 or NFYAv2 overexpressed NFYA-deficient SUM159 cells. All error bars represent SEM; n.s. not significant; *$P < 0.05$; **$P < 0.01$; ***$P < 0.001$.

Furthermore, we isolated breast cancer cells from these mice and analyzed the expression of these genes using western blotting. Similar to the results of immunohistochemical analysis, the expression of Acaca and Fasn was decreased in cells isolated from $Nfyav1^{-/-}$ breast cancer tissues. Simultaneously, the expression of Cpt1a, a critical rate-limiting enzyme of FAO, also decreased (Fig. 5j). Finally, we evaluated the expression of ACACA and FASN in tumour tissues obtained from xenograft experiments using human TNBC cells (Fig. 2c, f) using immunohistochemistry (Supplementary Fig. 10). The results showed that the tumour tissues derived from $NFYA^{+/+}$ SUM159 cells highly expressed these genes. In contrast, tumour tissues derived from $NFYA^{-/-}$ SUM159 cells showed reduced expression of these genes (Supplementary Fig. 10a). Furthermore, re-expression of NFYAv1 in $NFYA^{-/-}$ SUM159 cells restored the expression of these genes (Supplementary Fig. 10b). These results complement data from

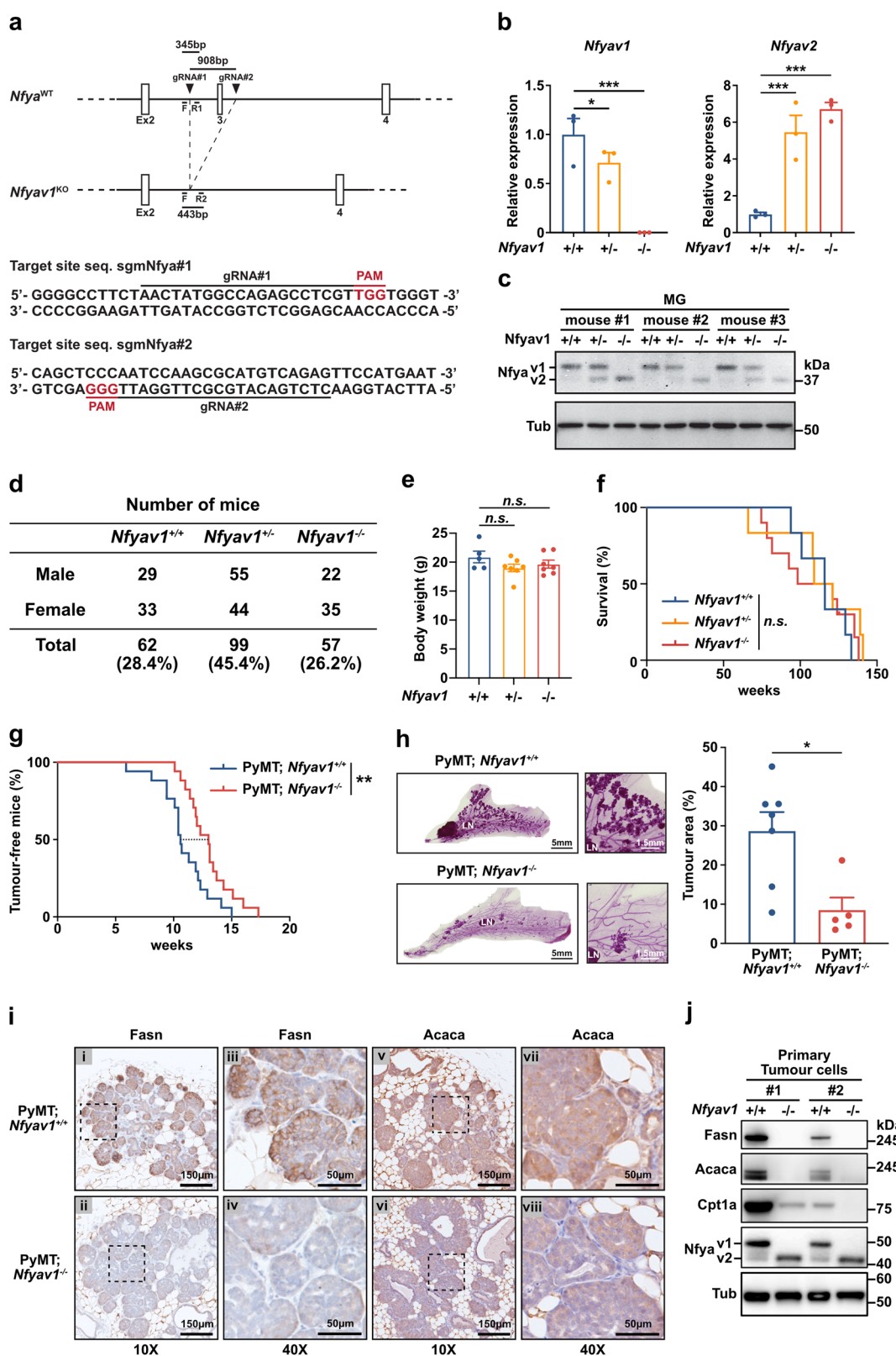

Nfyav1 knockout mice, in which NFYAv1 is a crucial regulator of lipogenic enzymes. Altogether, we showed that the loss of NFYAv1 causes a decrease in FAO by suppressing ACACA and FASN expression in vivo, thereby suppressing breast carcinogenesis, making NFYAv1 inhibition a safe and compelling candidate for TNBC therapeutic targets.

## Discussion

Here, we describe how NFYA regulates malignant behavior of tumour cells. Using in vivo and in vitro studies, we showed that NFYAv1 strongly supported TNBC cell growth and tumorigenesis. NFYAv1 could regulate de novo lipid synthesis and, through this regulation, promote the malignant behavior of cancer cells.

**Fig. 5 Nfyav1 enhances tumorigenesis via the regulation of ACACA and Fasn expression in vivo. a** Diagram of endogenous *Nfya* gene structure and Nfyav1 knockout structure. Using the CRISPR/Cas9 system with two gRNAs targeted intron 2 and 3, we deleted the exon 3 from genome DNA. **b, c** Confirming loss of Nfyav1 expression in mouse mammary epithelial cells. Littermate-controlled 8-weeks *Nfyav1*[+/+], *Nfyav1*[+/−], and *Nfyav1*[−/−] mice were analyzed by qRT-PCR (**b**) and western blot analysis (**c**) to qualify the expression of Nfyav1. N = 3 biologically independent experiments. All error bars represent SEM; *P < 0.05; ***P < 0.001. **d** Genotypes of offspring from *Nfyav1*[−/−] mouse intercrosses. **e** Body wight of mice at 8-week-old for *Nfyav1*[+/+], *Nfyav1*[+/−], and *Nfyav1*[−/−]. N = 5 for *Nfyav1*[+/+] mice, N = 7 for *Nfyav1*[+/−] and *Nfyav1*[−/−] mice biologically independent animals. **f** Kaplan–Meier survival curve for *Nfyav1*[+/+], *Nfyav1*[+/−], and *Nfyav1*[−/−]. N = 6 for *Nfyav1*[+/+] and *Nfyav1*[+/−] mice, N = 10 for *Nfyav1*[−/−] mice biologically independent animals. **g** Kaplan–Meier analysis of tumour-free mice for MMTV-PyMT; *Nfyav1*[+/+] (blue) and MMTV-PyMT; *Nfyav1*[−/−] (red). N = 17 biologically independent animals. **P < 0.01. **h** Representative images of whole-mount carmine alum staining of MMTV-PyMT; *Nfyav1*[+/+] or MMTV-PyMT; *Nfyav1*[−/−] mammary gland at 15-week-old. LN, lymph node. A bar graph shows the percentage of tumour area versus the total mammary fat pad in MMTV-PyMT; *Nfyav1*[+/+] or MMTV-PyMT; *Nfyav1*[−/−] mice. N = 7 for MMTV-PyMT; *Nfyav1*[+/+] mice, N = 5 for MMTV-PyMT; *Nfyav1*[−/−] mice biologically independent animals. Scale bars indicate 5 mm and 1.5 mm. *P < 0.05. **i** Breast cancer sections from both genotypes were stained with Fasn (panels i–iv) and Acaca (panels v–viii). Scale bars: 10× images, panels i, ii, v, vi, 150 μm; 40× images, panels iii, iv, vii, viii, 50 μm. **j** Protein expression analysis in PyMT-induced tumours from *Nfyav1*[+/+] or *Nfyav1*[−/−] mice.

---

This regulation was accomplished by directly promoting ACACA and FASN transcription.

Typically, lipogenesis pathway activity is maintained at a low level in normal tissues. However, in TNBC, high activity of lipogenesis has been reported to produce membrane phospholipids, signaling molecules, and energy for proliferation[10–12] (Supplementary Fig. 4a). Therefore, the lipogenic pathway is considered a potential therapeutic target for TNBC, for which no effective therapeutic targets have been identified. In preclinical models of TNBC, the inhibition of FASN alone or in combination with EGFR has been reported to have a practical antitumour effect on TNBC[43–45]. However, issues remain regarding the antitumour effects of blocking the lipogenesis pathway.

The first challenge is the need to select patients for whom lipogenesis blockade is effective, since FASN expression varies widely among TNBC patients causing differences in therapeutic efficacy[10,11]. Pathological examination showed that approximately 45% of the TNBC cases had high expression of FASN, whereas the remaining 55% had low or no expression of FASN[10,11]. Therefore, to obtain the appropriate therapeutic benefit of blocking the lipogenesis pathway, it is necessary to understand the regulatory mechanisms of lipogenesis and determine which patients may benefit from this treatment. Our analysis of NFYA splicing variant expression in breast cancer subtypes indicated a strong correlation between the expression pattern of NFYA splicing variants and breast cancer subtypes. In particular, a high NFYAv1/NFYAv2 ratio was characterized in Claudin[low] breast cancer subtype, which is highly aggressive and has a worse prognosis. Furthermore, we found that NFYAv1 promotes breast cancer malignancy in this subtype. It was also shown that this phenotype is because NFYAv1 promotes the transcription of lipogenic enzymes ACACA and FASN. In addition, these phenotypes in Claudin[low] breast cancer cells were consistent with the delayed breast cancer development phenotype in MMTV-PyMT; *Nfyav1*[−/−] mice. In short, the NFYAv1-lipogenesis axis enhances malignant behavior in the Claudin[low] breast cancer subtype. Disruption of this axis inhibits its malignant behavior, indicating that the NFYA expression pattern may be helpful as a marker to determine the therapeutic efficacy of lipogenesis pathway blockade. Previous studies have shown that splicing factors with EMT-dependent expression, ESRP, RBFOX2, and QKI, have been suggested to regulate the expression shift of EMT-dependent splicing variants of NFYA[46–48]. Therefore, further integrated analysis of the expression of these factors and NFYA splicing variants may be necessary to develop accurate markers for the therapeutic efficacy of lipogenesis pathway blockade.

The second challenge is clarifying whether lipogenesis pathway blockade has antitumour effects clinically. Zathy et al. showed

that the effect of FASN inhibition on cell proliferation in vitro does not necessarily translate into the same effect on antitumour efficacy in vivo[49,50]. In this study, we found that blockade of the lipogenesis pathway by NFYA deficiency suppresses the malignant behavior of Claudin[low] breast cancer cells. However, we also found that the addition of exogenous lipids restored this suppressed malignant behavior. Furthermore, the addition of exogenous lipids suppressed the binding of NFYA to the promoter regions of ACACA and FASN and suppressed the expression of these genes. Noteworthily, NFYA deficiency increased the expression of fatty acid translocase (CD36), a fatty acid transporter. These results suggest that the antitumour effect of lipogenesis pathway blockade is limited, and that cancer cells overcome the antitumour effect through exogenous lipid uptake when the lipogenesis pathway is blocked. Cancer-associated adipocytes, which exist in the tumour microenvironment, release free fatty acids into the tumour tissue, which are then taken up by cancer cells to promote cancer progression[51,52]. In short, the reason for the limited antitumour effect of the lipogenesis pathway blockade in vivo can be explained, at least in part, by the utilization of this mechanism by cancer cells. Nevertheless, several studies have confirmed the therapeutic effects of the FASN inhibitor[44,49,53,54]. In our in vivo mouse model, tumorigenesis was suppressed by the loss of the NFYAv1-lipogenesis axis. These results suggest that not all cancer cells can overcome the lipogenic pathway blockade by the uptake of exogenous lipids. Whether this difference is owing to subtype, cancer progression, or tumour microenvironment is currently unknown, and further research is needed. Currently, therapeutic strategies that inhibit CPT1A for FAO blockade are also being tested[55]. However, FAO blockade has little effect on the production of membrane phospholipids and signaling molecules, thus limiting the antitumour effect, and lipogenic blockade would be more advantageous. Indeed, in our study, the FASN inhibitor cerulenin suppressed cellular proliferation more than the CPT1A inhibitor etomoxir did. Dual-targeted therapy for prostate cancer with inhibition of de novo lipogenesis and uptake of fatty acids potently inhibits tumour growth compared with single-targeted therapy[56]. In the future, it will be necessary to consider a combined therapeutic strategy involving the lipogenesis pathway, lipid uptake, and FAO blockade for curing breast cancer.

Finally, the most serious challenge is its safety as a therapeutic target. Mice with knockout of the lipogenic enzyme Acly, Acaca, or Fasn exhibit embryonic lethality in early embryogenesis[15–17]. These phenotypes raise concerns regarding the safety of applying lipogenesis pathway blockade by inhibiting these genes as therapeutic targets. Nfya knockout mice are also reported to show embryonic lethality in early embryogenesis[40]. In the present study, we generated Nfyav1-specific knockout mice that were

born normal and showed no apparent morphological abnormalities or early onset of diseases. Given its safety and therapeutic efficacy, these results indicate that NFYAv1 may be a valuable and attractive therapeutic target for TNBC. Recently, inhibitors of NF-Y were identified[57,58]. These inhibitors bind to NF-Y and inhibit NF-Y-DNA complex formation, thereby blocking the transcriptional promotion by NF-Y. However, NFYA splicing variant-specific inhibitors have not been identified because the mechanism that induces the transcription of different target genes among NFYA splicing variants is still unclear. Further studies are necessary to elucidate the mechanism of transcriptional regulation by NFYA splicing variants.

This study revealed an essential role for NFYA in regulating lipid metabolism and the regulatory mechanisms of tumour malignant behavior by the NFYA-lipogenesis axis, demonstrating the importance of the lipid metabolic pathway as a therapeutic target for TNBC.

## Methods

**Generation of Nfyav1 knockout mice**. Electroporation introduced two single guide RNAs (sgRNA) targeting the intron 2 and 3 of Nfya (sgmNfya#1: 5′-AACT ATGGCCAGAGCCTCGT-3′ and sgmNfya#2: 5′-CTCTGACATGCGCTTGGA TT-3′) and human codon-optimized Cas9 into zygotes obtained from C57BL/6 J to remove exon 3 of Nfya (Fig. 5a). Treated zygotes were cultured overnight to the 2-cell stage and then transferred to pseudopregnant recipient mice. Founder mice were genotyped using primers (primer F: 5′-GAGTCCCAAGCCACTGATGA-3′, primer R1: 5′-ACCATGGATGAAGGAACTAGCC-3′, and primer R2: 5′-TCCT GCCTCCATATCCCAAC-3′) (Fig. 5a). To generate a PyMT-induced mouse breast cancer model, we obtained FVB/N-Tg(MMTV-PyVT)634Mul/J from The Jackson Laboratory (no. 002374) and backcrossed with C57BL/6 N females. After six generations, MMTV-PyMT mice were crossed with Nfyav1⁻/⁻ mice to generate MMTV-PyMT; Nfyav1⁻/⁻ mice.

**Animal experiments**. All mouse experiments were performed in accordance with Kyoto University, Kanazawa University, and Okayama University Institutional Animal Care and Use Committee approved protocol (Med Kyo13605, AP-153426, OKU-2018654, OKU-2018906). For orthotopic tumour growth assay, $1 \times 10^6$ cells were resuspended in 30 μl culture media containing 50 % Matrigel (Corning, no. 356230) and injected into the mammary fat pad of 6-week-old NOD-SCID female mice. Mice were sacrificed 50 days later to determine the tumour weight. For Nfyav1 knockout mice analysis including gene expression, body weight analysis, and mammary gland morphology, 8-week-old C57BL/6 N background Nfyav1⁺/⁺, Nfyav1⁺/⁻, and Nfyav1⁻/⁻ female mice were analyzed. For breast cancer tissue analysis, 15-week-old C57BL/6 N background MMTV-PyMT; Nfyav1⁺/⁺ and MMTV-PyMT; Nfyav1⁻/⁻ female mice were analyzed.

**Isolation of mouse tumour cells**. Primary mouse breast tumour cells were isolated from 18-week-old MMTV-PyMT female mice. Following harvest, tumours were cut into small pieces and digested for 1.5 h at 37 °C in 5 ml of DMEM/F12 containing 2 mg/ml of collagenase type I (WAKO, no. 037-17603) and 100 U/ml of hyaluronidase (Sigma, no. H3506). Following digestion, trypsin/EDTA was added, and the mixture was incubated for 5 min at 37 °C. Digested tumour samples were then pressed through 40-μm cell strainers. Finally, samples were centrifuged at 1000 rpm for 5 min at room temperature, resuspended in DMEM supplemented with 10% FBS and 1% penicillin and streptomycin, and plated on collagen type I coated 100 mm dishes (IWAKI, no. 4020-010-MYP).

**qRT-PCR analysis**. According to the manufacturer's instruction, total RNA was extracted from cells and mammary glands using TRIzol reagent (Thermo Fisher Scientific, no. 15596018). For qRT-PCR analysis, cDNA was synthesized from total RNA using a High-Capacity RNA-to-cDNA Kit (Applied Biosystems, no. 4387406). mRNA expression levels were determined by qRT-PCR with KAPA SYBR FAST qPCR Master Mix Kit (Kapa Biosystems, no. KK4610). Relative expression levels were normalized to human ACTB or mouse Actb. The primers used here are shown in Supplementary Data 1.

**Western blot analysis**. Total cellular extracts resolved by SDS-PAGE were transferred to PVDF membranes. Western blot was performed in TBST (100 mM Tris-HCl at pH7.5, 150 mM NaCl, 0.05% Tween-20) containing 5% BlockingOne (nacalai tesque, no. 03953-95). Immunoreactive protein bands were visualized using SuperSignal West Pico PLUS Chemiluminescent Substrate (Thermo Scientific, no. 34577).

**Cell culture**. All human breast cancer cells, NMuMG cells, and 293T cells were obtained from American Type Culture Collection. HMLE cells were obtained from R. Weinberg (MIT, USA). MDAMB231, NMuMG, and 293T cells were maintained in DMEM supplemented with 10% FBS. MCF7 cells were maintained in DMEM supplemented with 10% FBS and 10 μg/ml insulin. BT474, HCC1143, BT549, and MDAMB468 cells were maintained in RPMI1640 supplemented with 10% FBS. SUM149 and SUM159 cells were maintained in Ham's F12 supplemented with 5% FBS, 10 mM HEPES, 1 μg/ml Hydrocortisone, and 5 μg/ml Insulin. MCF10A and MCF12A cells were maintained in MEGM supplemented 100 ng/ml cholera toxin. HMLE cells were maintained in MEGM bullet kit (Lonza, no. CC-3150). All media were supplemented with 1% penicillin and streptomycin.

**CRISPR/Cas9 system**. CRISPR/Cas9 system was used to knockout the NFYA gene in SUM159, BT549, MCF7, and HMLE cells. NFYA and NFYAv1-specific gRNAs were determined using the candidates provided by GPP sgRNA Designer (https://portals.broadinstitute.org/gpp/public/analysis-tools/sgrna-design) and cloned into px459 vector (Addgene, no. 48139) for NFYA knockout and lentiCRISPRv2 vector (Addgene, no. 52961) for NFYAv1-specific knockout. Sequence for human NFYA sgRNA #1: GCCTTACCAGACAATTAACC, human NFYA sgRNA#2: GAGCA-GATTGTTGTCCAGGC, human NFYAv1 sgRNA#1: GCCCAGGTGGCATCCG CCTC, human NFYAv1 sgRNA#2: GGCCTGAGGCGGATGCCACC.

**Generation of retrovirus and lentivirus**. PCR products were then cloned into pMSCV-puro, pTRE3G-blast, or lentiCRISPRv2. For retrovirus preparation, 293 T cells were transfected with 300 ng of pMSCV-puro-NFYAv1-, -NFYAv2, -NFYAv1YA29mt, or -NFYAv2YA29mt together with 300 ng of packaging plasmid, pCL-Ampho. For lentivirus preparation, 293 T cells were transfected with 300 ng of pTRE3G-blast-SNAIL or lentiCRISPRv2-gRNA together with packaging plasmids (300 ng of pMDLg/pRRE, 150 ng of pMD2g, and 150 ng of pRSVRev). Cells were infected using the filtrated culture supernatant from 293T cells in the presence of 8 μg/ml polybrene.

**Cell proliferation**. For direct cell counting experiments, tumour cells were plated in triplicate at $0.2 \times 10^4$ cells per well of a 24-well plate. At indicated days, cells were trypsinized and counted.

For crystal violet staining experiments, tumour cells were plated $0.5 \times 10^4$ cells per well of a 24-well plate. At indicated days, cells were washed with PBS (-) and stained with 0.5% crystal violet solution for 20 min. The wells were washed with water, dried, and incubated with methanol for 20 min. The amount of dye reflecting the cell number was measured at 590 nm.

**Sphere formation assay**. Single-cell suspensions of cell lines were suspended at a density of 2000 cells/ml in DMEM/Ham's F12 (nacalai tesque, no. 11581-15) containing 20 ng/ml bFGF (Peprotech, no. 100-18B), 20 ng/ml EGF (Wako, no. 059-07873), 1× B27 (Thermo Scientific, no. 17504044), and 1% methylcellulose 400 (Wako, no. 132-05055) into 24-well ultra-low attachment plates (Corning, no. 3473). Spheres were counted after 7 days.

**Bodipy staining**. Cells were cultured on eight wells chamber slides with or without 2% lipid mixture for 3 days and fixed with 4% paraformaldehyde for 20 min at room temperature. After washing, cells were incubated with 3.8 μM of BODIPY 493/503 (Thermo Fisher Scientific, no. D3922) for 30 min at room temperature, then mounted with Vectashield hard-set mounting medium with DAPI (Vector laboratories, no. H-1500).

**Oxygen consumption rate (OCR) measurement**. In all, $5 \times 10^4$ cells were plated on XF24 cell culture plate (Agilent, no. 100777-004) with DMEM supplemented with 1% FBS, 0.5 mM glucose, 1 mM Glutamax, 0.5 mM Carnitine, 10 mM HEPES, 1 μg/ml Hydrocortisone, and 5 μg/ml Insulin. For measuring OCR in response to lipid stimulation, cells were cultured with 33 μM Palmitate-BSA or free fatty acid reduced-BSA in Krebs-Henseleit buffer supplemented with 2.5 mM Glucose, 0.5 mM Carnitine, 5 mM HEPES, 1 μg/ml Hydrocortisone, and 5 μg/ml Insulin at 37 °C for 1 h without $CO_2$ control. 1 μM Oligomycin A (Cell Signaling Technology, no. 9996) was injected to inhibit ATPase V. Maximal OCR was induced by exposing cells to mitochondrial uncoupler, 2 μM FCCP (Sigma-Aldrich, no. C2920). 1 μM Antimycin A (Sigma-Aldrich, no. A8674) and 1 μM rotenone (Sigma-Aldrich, no. R8875) were added to disrupt all mitochondria-dependent respiration. XF24 was used to measure OCR under a 3 min period protocol, followed by 2 min mixing and 3 min incubation.

**CUT & RUN analysis**. CUT & RUN experiments were performed using $3 \times 10^5$ cells with the CUT&RUN assay kit (CST, no. 86652). In brief, cells were washed, bound to activated Concanavalin A magnetic beads, and permeabilized with antibody binding buffer containing Digitonin. The bead-cell complex was incubated overnight with 0.7 μg of NFYA monoclonal antibody at 4 °C. The bead-cell complex was washed with Digitonin buffer and incubated pAG-MNase solution for 1 h at 4 °C. After washing with Digitonin buffer, 2 mM calcium chloride was added to activate pAG-MNase and incubated for 30 min at 4 °C. After incubation, the

reaction was stopped with stop buffer. DNA fragments were released by incubation for 10 min at 37 °C and purified with Fast Gene Gel/PCR extraction kit (Nippon genetics, no. FG-91202). The DNA fragments were quantified by qRT-PCR analysis. The primers used here are shown in Supplementary Data 1.

**Histology and immunohistochemistry (IHC).** For IHC, paraffin sections were deparaffinized, dehydrated, and subjected to heat-induced antigen retrieval in a pressure cooker using 10× G-Active pH6 (Geno Staff, no. ARSC6-01). Slides were incubated for 10 min with 3% $H_2O_2$, blocked for 1 h with BlockingOne Histo (nacalai tesque, no. 06349), and incubated with primary antibody overnight at 4 °C in Can Get Signal immunostain solution A (TOYOBO, no. NKB-501). Signals were enhanced using Vectastain ABC Elite kit (Vector Laboratories, no. PK-6101) and visualized by ImmPACT DAB (Vector Laboratories, no. SK-4105) and counter-staining with Mayer's Hematoxylin (Wako, no. 131-09665).

For immunofluorescence staining, paraffin sections were deparaffinized, dehydrated, and subjected to heat-induced antigen retrieval in a pressure cooker using 10× G-Active pH6. Slides were blocked for 1 hour with BlockingOne Histo and incubated with primary antibody overnight at 4 °C in 5% BlockingOne Histo in PBS (−). The next day, the slides were labeled with Alexa Fluor 568 (1:500; Thermo Fisher Scientific, no. A-11036) or Alexa Fluor 488 (1:500; Thermo Fisher Scientific, no. A-11001) and mounted the slides with Vectashield hard-set mounting medium with DAPI (Vector Laboratories, no. H-1500).

For whole-mount carmine alum staining, mammary fat pads, including mammary gland, were fixed on glass slides with fixation buffer (25% acetic acid, 75% ethanol) overnight. Tissues were hydrated and stained with carmine alum solution (Stemcell Technologies, no. 07070). Tissues were dehydrated, cleared in Histoclear (National Diagnostics, no. HS-200), and mounted with Permount (Falma, no. SP15-100-1).

**Antibodies.** The following monoclonal (mAb) and polyclonal (pAb) primary antibodies were used for western blot and IHC: NFYA pAb (1:500; Santa Cruz, no. sc-10779), NFYA mAb (1:500; Santa Cruz, no. sc-17753), E-Cadherin mAb (1:1000; BD, no. 610181), Vimentin mAb (1:1000; BD, no. 550513), SNAI 1 mAb (1:500; Santa Cruz, no. sc-28199), Flag mAb (1:1000; Sigma, no. F3165), FASN Rabbit mAb (1:1000 for western blot, 1:100 for IHC; CST, no. 3180), ACACA Rabbit mAb (1:1000 for western blot, 1:100 for IHC; CST, no. 3676), CPT1A Rabbit mAb (1:1000; Abcam, no. Ab234111), ACADVL mAb (1:1000; Santa Cruz, no. sc-376239), SREBP1 mAb (1:500; Santa Cruz, no. sc-13551), CD36 pAb (1:500; Santa Cruz, no. sc-9154), Keratin 14 pAb (1:500; BioLegend, no. 905304), Keratin 8 mAb (1:500; BioLegend, no. 904804), and α-Tubulin mAb (1:2000; Sigma, no. T5168).

**Reagents.** The following reagents were used: Forskolin (10 μM; Wako, no. 067-02191), Lipid mixture 1 (2%; Sigma, no. L0288), Etomoxir (3 μM; Abcam, no. Ab254445), Betulin (5 μg/mL; Wako, no. 020-17371), and Cerulenin (20 μM; Biolinks, no. BLK-0380).

**Statistics and reproducibility.** All experiments were conducted in series of three or more, and the number used in each experiment is indicated in the figure legend. Sample sizes were determined based on previous studies using similar experiments. Sample sizes for all in vitro experiments were determined based on previous studies using similar experiments. We determined the sample size for animal experiments based on pilot experiments. Data are presented as the mean ± SEM. The statistical significance of the difference between experimental groups was assessed using an unpaired two-tailed Student's $t$ test using GraphPad Prism 9 software. $P$ values of <0.05 were considered significant. All experiments were repeated at least twice independently to ensure reproducibility.

**Reporting summary.** Further information on research design is available in the Nature Portfolio Reporting Summary linked to this article.

## Data availability

The authors declare that all data discussed in the paper will be available to the readers. All uncropped images of western blot analysis are available in Supplementary Fig. 11. The numerical source data are provided in Supplementary Data 2. Newly generated plasmids are available from Addgene (Addgene ID: 202633-202640). All other data are available from the corresponding author upon reasonable request.

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

## Acknowledgements

We thank members of Medical Protein Engineering Laboratory, Okayama University, Division of Oncology and Molecular Biology, Kanazawa University, and DSK project, Kyoto University for their help and input. Particularly, we thank Mr. R. Ito, Mr. H. Sakai, and Dr. S. Ikeda for technical assistance and stimulating discussion. We would like to thank Editage (www.editage.com) for English language editing. This work was supported by JSPS KAKENHI Grant Numbers JP22K07152 (to N.O.), JP19K07640 (to N.O.), JP15K18407 (to N.O.), JSPS KAKENHI Grant Number JP16H06276 (AdAMS) (to N.O.), Cancer Research Institute of Kanazawa University (to N.O.), Takeda Science Foundation (to N.O.), Wesco Scientific Promotion Foundation (to N.O.), and The Okayama Foundation for Science and Technology (to N.O.).

## Author contributions

N.O. conceived the project and designed experiments. N.O., C.U., M.S., and G.T. performed experiments and analyzed data. S.K. and C.T. performed metabolic analysis and provided intellectual input on metabolic mechanisms. H.M. provided technical help to generate mice. K.Y. performed experiments and provided intellectual input on breast cancer. N.O. and C.T. wrote and edited the manuscript. All authors provided comments on the manuscript and gave final approval.

## Competing interests

The authors declare no competing interests.
