## [Peer Review File · Communications Biology]

Reviewers' comments:

Reviewer #1 (Remarks to the Author):

The manuscript by Okada and colleagues shows that the NFYA1 splice variant is overexpressed in basal breast cancers, and drives a lipogenic phenotype, where cells upregulate proteins involved with de novo lipogenesis. The manuscript is well written and logically addresses pertinent questions around the biology of NFYA1 and regulation of metabolism. I do have some suggestions that I think should be addressed prior to publication.

The authors predominantly use one cell line throughout the manuscript - SUM159 – thus there is a question whether some of these effects are cell line specific, and dependent on the specific epistatic interactions present in this particularly TNBC line. Ideally, I would like to see another TNBC cell line where NFYA1 has been specifically deleted and the same phenotypes are observed.

The authors correctly discuss the cooperative regulation of lipogenic genes with the master regulators SREBPs, yet there is limited evidence that in their system SREBP activity is indeed increased when NFYA1 is lost. I would suggest that some experiments should be performed that confirm this interaction in their models i.e.

- inhibiting SREBP1 activity with betulinic acid and assessing the impact on lipid genes
- Using a sterol regulatory element reporter assay to show SREBP is more active when NFYA1 is expressed
- Assess SREBP protein cleavage and nuclear localisation
- ChIP SREBP at the promotor of FASN (and other genes) and showing that this is dependent on NFYA1 status

The authors have powerful in vivo data, thus it would be really impactful to assess the histological expression of lipid enzymes in these all tumour models, including the SUM159 xenografts. This would show that under in vivo conditions NFYA1 loss or expression is a major regulator of lipid enzymes, and another factor does not compensate and influence their expression. This would complement the knockout model data nicely.

The authors posit that the increased dependency on glucose means more of this carbon is being shuttled into lipids, but this does not have to be the case because carbon can come from multiple other sources i.e. glutamine, acetate, lactate, fatty acid oxidation (or a combination of all of the above). The authors would need to perform fluxomics using a glucose tracer, showing that carbon is reduced into palmitate to support their claims.

Minor comments

"Tumor" should be "tumour"

In vivo and in vitro should be in italics

Reviewer #2 (Remarks to the Author):

The authors of this manuscript studied function of NF-Y transcription factor in triple-negative breast cancer in the context of regulation of lipid metabolism. Overall, the experimental design is well-done, which specifically addressed role of NF-YA1 variant (long-variant) in breast tumor biology using both human cell lines and mouse models. Interestingly, the NF-YA1 mouse model reported in the study is

not embryonic lethal due to expression of shorter NFYAv2 variant (containing DNA-binding domain); the mouse model greatly helped to address specific role of NF-Yv1 in mouse breast cancer models and in lipid metabolism. The mouse model can be also a valuable resource in future to study NFYAv1 function in different cancer models.

In my view, the manuscript should be accepted for publication after the authors address few minor comments as described below.

1. I suggest the title should be "NFYA Variant" instead of "NFYA"

2. As shown in Fig. 5, deletion of exon 3 of mouse NFYA gene resulted inhibition of NFYAV1 expression but increased NFYAV2 expression, which contains both DNA binding and subunit interaction with NFYB/NFYC heterodimer domains, meaning NFYAV2 can form functional NFY-DNA complex. Although mechanistic analysis of the observation is not within the scope of current manuscript, it can however initiate future study since NF-Y regulates many mammalian genes associated with cell proliferation and differentiation in addition to lipid metabolism. The authors can clarify their observation in this respect.

Reviewer #3 (Remarks to the Author):

The manuscript entitled "NFYA promotes the malignant behavior of triple negative breast cancer through the regulation of lipid metabolism" by Okada et al is a good piece of work connecting molecular mechanism of NFYA-mediated lipogenesis in the progression of triple negative breast cancer. The authors have carried out extensive experiments to support their findings. However there are a few concerns that need further clarifications:

1. The authors have used a wide panel of breast cancer cell lines of varying receptor status to show the relative expression of NF-YA variants and EMT markers. In figure 1e, it was really interesting to observe a correlation in NF-YAv1 expression and E-cadherin in MCF-7 and BT474 receptor positive cells. On the contrary in TNBC cells (claudin low), we observe an increase in mesenchymal marker vimentin. But it is really surprising to notice that the authors have not included MD-AMB-468 in the panel of Figure 1d and 1e, although they have used MDA-MB-468 for all the other experiments in results 1. It is also noticed that in Figure 1f, the MDA-MB-468 cells without treatment with Dox showed a high expression of E-cadherin. This is unexpected, as MDA-MB-468 being TNBC should show relatively high expression of mesenchymal marker vimentin (similar to MD-AMB-231 Figure 1e). Hence provide suitable explanation to prove the hypothesis.

2. In lines 134-136, the authors mention about phenotypic switching between variants NF-YAv1 and NF-YAv2 with induction of SNAIL. Doxycycline treatment is used as Tet inducer, it would be appropriate to check the expression of SNAIL by Western blotting to support Figure 1f.

3. There are opposing roles of E-cadherin in breast cancer (Kowalski, P.J., Rubin, M.A. & Kleer, C.G. E-cadherin expression in primary carcinomas of the breast and its distant metastases. *Breast Cancer Res* 5, R217 (2003) <https://doi.org/10.1186/bcr651>). The authors need to provide suitable explanation to substantiate their hypothesis of NF-YA variants and EMT markers in breast cancer.

4. In lines 137-139 the authors mention about use of forskolin to reverse the expression from mesenchymal to epithelial. But the supplementary figure 1f showed no significant change in the expression. Also in supplementary figure 1g, there is no difference in the expression of E-cadherin and vimentin among NF-YA+/+ NF-YA+/- and NF-YA-/-. The difference in expression is only observed in the presence or absence of SNAIL, so it is not NF-YA which is controlling EMT rather external factors present in the tumor microenvironment control the phenotypic switching. At the end of result 1 the authors mention that NF-YA by itself is not an inducer of EMT. What would be the underlying

mechanism, provide explanation for the same.

5. In figure 2c, the tumor mass of NF-YA+/- is not shown, does it have a normal tumor weight comparable to wild type? Also, the graphical representation in figure 2c had two panels of NF-YA-/- .Please make sure it is correct.

6. In figure 2f, NFYA^{v2} transfection also increased tumor size compared to mock, the authors have not provided suitable explanation for this observation. Also, the graphical representation of tumor weight of NFYA^{v1} showed wide difference (figure 2f), what could be the reason. It is recommended to plot tumor volume rather than tumor weight.

7. In supplementary figure 3c, we find that FASN inhibitor cerulenin alone is sufficient to inhibit sphere forming ability irrespective of the presence or absence of NFYA. The authors claim that NFYA is responsible for turning on lipogenic genes like FASN and others. This concept is to be clarified.

8. In results 3, the authors elaborate the role of Palmitic acid in increasing OCR in Wild type and KO cells. However, the lines 203-205 is not convincing, please make the statements clear for the readers.

8. In lines 215-216, the authors claim that NFYA deficient cells are more sensitive to glucose deprivation than Wild type cells. The authors cannot make such a direct correlation with lipid metabolism from this qualitative experiment alone. It is hence recommended to check the levels of lipogenic genes upon glucose starvation in NFYA mutant cells to draw such a conclusion.

9. In lines 235-237, the immunofluorescence explanation on FASN in NFYA -/- supplemented with supplementary Figure 4a is of very low resolution. Replace the image to substantiate the theory.

10. In line 241-243, the authors claim that Etomoxir (ETO) increases CPT1A expression. This in turn is highly contradictory as ETO has been shown to be an inhibitor of CPT1 in line 218. Similarly cerulenin a reported inhibitor of FASN was used for the experiment ,but the western blot images provided in Supplementary figure 4d donot show a significant down-regulation of FASN protein expression. Provide suitable explanation.

11. Provide an appropriate reasoning for the expression of CD36 expression (supplementary figure 4f) in lines 263-267.

The authors have not elaborated the purpose of using various cell lines and compounds in detail. It makes the idea complicated for readers. So it is recommended to improve the language of the manuscript. There are a few grammatical errors which need to be rectified. References need to be made uniform.

COMMSBIO-22-1902-T

NFYA promotes the malignant behavior of triple-negative breast cancer through the regulation of lipid metabolism

Okada *et al.*

We thank the reviewers for carefully and critically evaluating our manuscript. The helpful comments and suggestions provided by the reviewers have aided in further improving the quality of our study. Below are the reviewers' comments in black and our corresponding responses in blue.

Response to Reviewer #1:

The manuscript by Okada and colleagues shows that the NFYAv1 splice variant is overexpressed in basal breast cancers, and drives a lipogenic phenotype, where cells upregulate proteins involved with de novo lipogenesis. The manuscript is well written and logically addresses pertinent questions around the biology of NFYAv1 and regulation of metabolism. I do have some suggestions that I think should be addressed prior to publication.

1. The authors predominantly use one cell line throughout the manuscript - SUM159 – thus there is a question whether some of these effects are cell line specific, and dependent on the specific epistatic interactions present in this particularly TNBC line. Ideally, I would like to see another TNBC cell line where NFYAv1 has been specifically deleted and the same phenotypes are observed.

We thank the reviewer for this constructive suggestion. As suggested, we found that in another NFYA-deficient TNBC cell line generated using BT549 cells, NFYA deficiency suppressed the ability of cells to proliferate and form spheres. Moreover, the lipid mixture in the culture medium restored cell growth and the sphere-forming ability of NFYA-deficient BT549 cells. We also detected that NFYA deficiency in BT549 cells decreased the expression of ACACA, FASN, and CPT1A. These phenotypes are consistent with those of NFYA-deficient SUM159 cells, indicating that the effects of NFYA are universal to TNBCs rather than cell-line specific. These results are included in the revised manuscript with the following text and figures.

In lines 198-204:

“To confirm that the restoration of malignant behavior by lipid addition is not a cell line-specific effect but a general effect on TNBCs, we generated another NFYA-deficient TNBC cell line using BT549 cells and confirmed their malignant behavior. The results showed that NFYA-deficient BT549 cells exhibited phenotypes identical to those of NFYA-deficient SUM159 cells, indicating that these phenotypes are not cell line-specific but are general to TNBCs (Supplementary Fig. 5a, b).”

Reviewer Figure 1 (Supplementary Fig. 5a, b; Supplementary Fig. 6c)

a Cumulative population of cells was measured every other day for 5 days in NFYA^{+/+} and NFYA^{-/-} BT549 cells with or without the addition of lipid mixture. **b** Representative images of sphere formation by NFYA^{+/+} and NFYA^{-/-} BT549 cells with or without the addition of lipid mixture. A bar graph shows the number of spheres larger than 60 μm in each group. **c** Western blot analysis of the expression levels of differentially expressed genes in NFYA^{+/+} and NFYA^{-/-} BT549 cells treated with or without lipid mixture.

2. The authors correctly discuss the cooperative regulation of lipogenic genes with the master regulators SREBPs, yet there is limited evidence that in their system SREBP activity is indeed increased when NFYA^{v1} is lost. I would suggest that some experiments should be performed that confirm this interaction in their models.

i.e.

- inhibiting SREBP1 activity with betulinic acid and assessing the impact on lipid genes
- Using a sterol regulatory element reporter assay to show SREBP is more active when NFYA^{v1} is expressed
- Assess SREBP protein cleavage and nuclear localisation
- ChIP SREBP at the promotor of FASN (and other genes) and showing that this is dependent on NFYA^{v1} status

We thank the reviewer for the insightful comment. To investigate the cooperative effects between NFYA and SREBP1, we first examined the status of SREBP1 in *NFYA*^{+/+} and *NFYA*^{-/-} SUM159 cells. Surprisingly, we observed that the nuclear form of SREBP1 was significantly enhanced in *NFYA*^{-/-} cells compared to *NFYA*^{+/+} cells. Treatment with the SREBP1 inhibitor, betulin, eliminated the nuclear form of SREBP1. These results indicate that NFYA deficiency increases SREBP1 activity. Furthermore, the effects of these conditions on lipogenic enzymes, ACACA and FASN, were examined, and the results showed that either NFYA deficiency or SREBP1 inhibition reduced the expression of these enzymes. These results indicate that NFYA and SREBP1 cooperatively regulate the expression of these enzymes and that dysfunction of either NFYA or SREBP1 results in dysregulation of their expression. These results are included in the revised manuscript with the following text and figures.

In line 287-301:

“NFYA cooperatively regulates the expression of ACACA and FASN with SREBP1. Although systematic global gene expression analysis has shown that NF-Y and SREBP1 cooperatively regulate *de novo* lipogenesis, the interaction between NF-Y and SREBP1 remains unclear²⁵. To elucidate this interaction, we evaluated SREBP1 expression in NFYA-deficient cells. We observed increased transcription of *SREBF1* encoding SREBP1 in NFYA-deficient cells compared to that in NFYA wild-type cells, and a marked enhancement of the nuclear form of SREBP1 (Supplementary Fig. 8a, c). Furthermore, analysis of lipogenic enzyme expression under these conditions revealed decreased expression of ACACA and FASN, despite the increased activity of SREBP1 (Supplementary Fig. 8b, c). Treatment with SREBP1 inhibitor betulin also decreased the expression of these enzymes in the presence of NFYA (Supplementary Fig. 8b, c). These results indicate that NFYA and SREBP1 cooperatively regulate the expression of these enzymes and that dysfunction of either NFYA or SREBP1 results in deregulation of the expression of these enzymes.”

Reviewer Figure 2 (Supplementary Fig. 8a-c)

a Western blot analysis of the expression levels of SREBP1 in *NFYA*^{+/+} and *NFYA*^{-/-} SUM159 cells treated with or without betulin or lipid mixture. fl-SREBP1; full-length SREBP1, n-SREBP1; nuclear SREBP1. Asterisk indicates nonspecific bands. **b**, **c** Western blot (**b**) and qRT-PCR analysis (**c**) of the expression levels of differentially expressed genes in *NFYA*^{+/+} and *NFYA*^{-/-} SUM159 cells treated with or without betulin.

3. The authors have powerful in vivo data, thus it would be really impactful to assess the histological expression of lipid enzymes in these all tumour models, including the SUM159 xenografts. This would show that under in vivo conditions NFYA v1 loss or expression is a major regulator of lipid enzymes, and another factor does not compensate and influence their expression. This would complement the knockout model data nicely.

We thank the reviewer for this constructive suggestion. As suggested, we performed immunohistochemistry for ACACA and FASN in tumour tissues derived from xenograft experiments. The results showed that tumour tissues derived from *NFYA*^{+/+} SUM159 cells highly expressed those genes. In contrast, tumour tissues derived from *NFYA*^{-/-} SUM159 cells significantly reduced the expression of those genes. Furthermore, the re-expression of NFYAv1 in *NFYA*^{-/-} SUM159 cells restored the expression of those genes. These results complement data from Nfyav1 knockout mice, in which NFYAv1 is a critical regulator of lipogenic enzymes. These results are included in the revised manuscript with the following text and figures.

In line 349-362:

“Finally, we evaluated the expression of ACACA and FASN in tumour tissues obtained from xenograft experiments using human TNBC cells (Fig. 2c, f) using immunohistochemistry (Supplementary Fig. 10). The results showed that the tumour tissues derived from *NFYA*^{+/+} SUM159 cells highly expressed these genes. In contrast, tumour tissues derived from *NFYA*^{-/-} SUM159 cells showed significantly reduced expression of these genes (Supplementary Fig. 10a). Furthermore, re-expression of NFYAv1 in *NFYA*^{-/-} SUM159 cells restored the expression of these genes (Supplementary Fig. 10b). These results complement data from Nfyav1 knockout mice, in which NFYAv1 is a crucial regulator of lipogenic enzymes. Altogether, we showed that the loss of NFYAv1 causes a decrease in FAO by suppressing ACACA and FASN expression *in vivo*, thereby suppressing breast carcinogenesis, making NFYAv1 inhibition a safe and compelling candidate for TNBC therapeutic targets.”

Reviewer Figure 3 (Supplementary Fig. 10a, b)

a Tumour tissue sections derived from *NFYA*^{+/+}, *NFYA*^{+/-}, and *NFYA*^{-/-} SUM159 cells were stained with FASN (panels i-vi) and ACACA (panels vii-xii). **b** Tumour tissue

sections derived from *NFYA*^{-/-} SUM159 cells overexpressed each variant of *NFYA* were stained with FASN (panels i-vi) and ACACA (panels vii-xii).

4. The authors posit that the increased dependency on glucose means more of this carbon is being shuttled into lipids, but this does not have to be the case because carbon can come from multiple other sources i.e. glutamine, acetate, lactate, fatty acid oxidation (or a combination of all of the above). The authors would need to perform fluxomics using a glucose tracer, showing that carbon is reduced into palmitate to support their claims.

We thank the reviewer for the critical comment. We agree with the reviewer that our results alone cannot show a direct correlation with glucose metabolism and that this is an overstatement. Since a direct correlation with glucose metabolism is beyond the scope of this paper, we have removed the results of Supplementary Fig. 3g from this manuscript. We intend to clarify them through future studies.

5. "Tumor" should be "tumour". In vivo and in vitro should be in italics

We thank the reviewer for the comment. As suggested, we corrected all "tumor" to "tumour" and "in vivo" and "in vitro" to italic.

Response to Reviewer #2:

The authors of this manuscript studied function of NF-Y transcription factor in triple-negative breast cancer in the context of regulation of lipid metabolism. Overall, the experimental design is well-done, which specifically addressed role of NF-YAv1 variant (long-variant) in breast tumor biology using both human cell lines and mouse models. Interestingly, the NF-YAv1 mouse model reported in the study is not embryonic lethal due to expression of shorter NF-YAv2 variant (containing DNA-binding domain); the mouse model greatly helped to address specific role of NF-YAv1 in mouse breast cancer models and in lipid metabolism. The mouse model can be also a valuable resource in future to study NF-YAv1 function in different cancer models.

In my view, the manuscript should be accepted for publication after the authors address few minor comments as described below.

1. I suggest the title should be “NFYA Variant” instead of “NFYA”.

We thank the reviewer for the comment. As suggested, we revised the title as follows.

“NFYA variant promotes the malignant behavior of triple-negative breast cancer through lipid metabolism regulation”

2. As shown in Fig. 5, deletion of exon 3 of mouse NFYA gene resulted inhibition of NFYAV1 expression but increased NFYAV2 expression, which contains both DNA binding and subunit interaction with NFYB/NFYC heterodimer domains, meaning NFYAV2 can form functional NFY-DNA complex. Although mechanistic analysis of the observation is not within the scope of current manuscript, it can however initiate future study since NF-Y regulates many mammalian genes associated with cell proliferation and differentiation in addition to lipid metabolism. The authors can clarify their observation in this respect.

We thank the reviewer for the comment. We agree with the reviewer that NF-Y regulates the expression of many genes and differentially regulates transcription among NFYA splicing variants. These need to be comprehensively clarified in the future. However, these analyses are outside the scope of this paper, and we hope to clarify them in future studies, and we have added the following text to the discussion part.

In line 456-459:

“However, NFYA splicing variant-specific inhibitors have not been identified because the mechanism that induces the transcription of different target genes among NFYA splicing variants is still unclear. Further studies are necessary to elucidate the mechanism of transcriptional regulation by NFYA splicing variants.”

Response to Reviewer #3:

The manuscript entitled “NFYA promotes the malignant behavior of triple negative breast cancer through the regulation of lipid metabolism” by Okada et al is a good piece of work connecting molecular mechanism of NFYA-mediated lipogenesis in the progression of triple negative breast cancer. The authors have carried out extensive experiments to support their findings. However there are a few concerns that need

further clarifications:

1. The authors have used a wide panel of breast cancer cell lines of varying receptor status to show the relative expression of NF-YA variants and EMT markers. In figure 1e, it was really interesting to observe a correlation in NF-YAv1 expression and E-cadherin in MCF-7 and BT474 receptor positive cells. On the contrary in TNBC cells (claudin low), we observe an increase in mesenchymal marker vimentin. But it is really surprising to notice that the authors have not included MD-AMB-468 in the panel of Figure 1d and 1e, although they have used MDA-MB-468 for all the other experiments in results 1. It is also noticed that in Figure 1f, the MDA-MB-468 cells without treatment with Dox showed a high expression of E-cadherin. This is unexpected, as MDA-MB-468 being TNBC should show relatively high expression of mesenchymal marker vimentin (similar to MDAMB-231 Figure 1e). Hence provide suitable explanation to prove the hypothesis.

We thank the reviewer for the comment. TNBC cells have been subtype-classified using indicators such as gene expression profiles, epigenomic alterations, and immune responsiveness. Among them, classification based on EMT molecular profiles as an indicator has been reported to correlate strongly with patient prognosis and resistance to therapy. Our study focused on mesenchymal TNBC subtype (MDA-MB-231, BT549, and SUM159 cells), which represent the majority of TNBC cells, are highly malignant, and have no identified therapeutic targets. There is also epithelial TNBC subtype such as MDA-MB-468 cells, but these are relatively mild malignancies and have shown therapeutic efficacy with drugs targeting DNA repair enzymes such as PARP inhibitors. Therefore, this study did not focus on epithelial TNBC subtype cells (MDA-MB-468 cells) but only to confirm that EMT also switches the expression of the NFYA splicing variants in TNBC cells as well. We have added the following text for clarity to the reader.

In line 47-50:

“TNBCs are classified into four subtypes according to their genetic profile: basal-like immunosuppressed, basal-like immune-activated, mesenchymal, and luminal androgen receptor². In particular, the mesenchymal TNBC subtype is associated with a worse patient prognosis³⁻⁶.”

2. In lines 134-136, the authors mention about phenotypic switching between variants NF-YAv1 and NF-YAv2 with induction of SNAIL. Doxycycline treatment is used as Tet inducer, it would be appropriate to check the expression of SNAIL by Western blotting to support Figure 1f.

We thank the reviewer for the comment. As suggested, we detected SNAIL expression by western blot analysis. The results showed that SNAIL was expressed from day 1 after doxycycline treatment.

Reviewer Figure 4 (Fig. 1f)

a MDA-MB-468 cells were infected with lentivirus to express tet-on SNAIL and induced SNAIL expression by 1 µg/ml of doxycycline (Dox). Western blot analysis of NFYA protein level in the cells from day 0 to day 7 post-induction. E-CAD and VIM are markers for epithelial and mesenchymal cells, respectively.

3. There are opposing roles of E-cadherin in breast cancer (Kowalski, P.J., Rubin, M.A. & Kleer, C.G. E-cadherin expression in primary carcinomas of the breast and its distant metastases. Breast Cancer Res 5, R217 (2003) <https://doi.org/10.1186/bcr651>). The authors need to provide suitable explanation to substantiate their hypothesis of NF-YA variants and EMT markers in breast cancer.

We agree that the role of E-cadherin in metastasis remains controversial. However, the EMT program has been reported to be involved in malignant tumor progression as well as metastasis. In breast cancer, the expression of characteristic mesenchymal genes has also been reported to be generally enriched in basal and TNBC subtypes. Therefore, we used Vimentin and E-cadherin expression to classify breast cancer subtypes correlated with the malignancy, which correlates with the expression of NFYA

splicing variants.

4. In lines 137-139 the authors mention about use of forskolin to reverse the expression from mesenchymal to epithelial. But the supplementary figure 1f showed no significant change in the expression. Also in supplementary figure 1g, there is no difference in the expression of E-cadhehin and vimentin among NF-YA+/+ NF-YA+/- and NF-YA-/. The difference in expression is only observed in the presence or absence of SNAIL, so it is not NF-YA which is controlling EMT rather external factors present in the tumor microenvironment control the phenotypic switching. At the end of result 1 the authors mention that NF-YA by itself is not an inducer of EMT. What would be the underlying mechanism, provide explanation for the same.

We thank the reviewer for this constructive comment. Our results in Supplementary Fig. 1f indicate that forskolin treatment for 15 days reduced the expression of the mesenchymal marker VIM by 38%. However, the expression of epithelial marker E-CAD remained at undetectable levels. These indicate that forskolin treatment induces mesenchymal-to-epithelial transition (MET), but its effects are limited and do not lead to complete MET induction. However, despite the limited MET, it shows a 33% decrease in NFYAv1 expression and a 22% increase in NFYAv2 expression, indicating that the NFYA splicing variants expression shift is an epithelial-mesenchymal type-dependent reversible response.

In addition, our results in Supplementary Fig. 1g show that loss of NFYA did not affect SNAIL-induced EMT in human mammary epithelial cells. These results indicate that NFYA is not an EMT regulator but a factor that functions through an EMT-dependent splicing variant expression shift.

Previous studies have suggested that splicing factors, ESRP, RBFOX2, and QKI, regulate EMT-dependent NFYA splicing variants shift (Zong *et al.*, 2014; Nieto *et al.*, 2016; Yang *et al.*, 2016). In epithelial cells, the epithelial cell-specific splicing factor, ESRP, promotes exon 3 skipping by binding to the upstream intron. In mesenchymal cells, the mesenchymal cell-specific splicing factors, RBFOX2 and QKI, promote exon 3 inclusion by binding to the downstream intron. These are suggested to achieve an EMT-dependent shift of NFYA splicing variant expression. We have revised the text as follows.

In line 139-151:

“Next, we examined whether the shift in NFYA expression is a reversible reaction. SUM159 cells were treated with forskolin, which is known to induce mesenchymal-epithelial transition (MET)³². We found that forskolin treatment decreased the expression of the mesenchymal marker vimentin, whereas the expression of the epithelial marker E-cadherin remained undetectable. These results indicate that forskolin treatment induces MET; however, its partial effect does not lead to complete MET induction. Despite partial MET induction, the ratio of NFYAv2 to NFYAv1 increased, indicating that the switching of NFYA variant expression is reversible (Supplementary Fig. 1f). In addition, NFYA deficiency in HMLE cells did not affect SNAIL-induced EMT progression, confirming that NFYA is not an EMT regulator but a factor that functions by shifting EMT-dependent splicing variant expression (Supplementary Fig. 1g).”

In line 401-407:

“Previous studies have shown that splicing factors with EMT-dependent expression, ESRP, RBFOX2, and QKI, have been suggested to regulate the expression shift of EMT-dependent splicing variants of NFYA⁴⁶⁻⁴⁸. Therefore, further integrated analysis of the expression of these factors and NFYA splicing variants may be necessary to develop accurate markers for the therapeutic efficacy of lipogenesis pathway blockade.”

5. In figure 2c, the tumor mass of NF-YA+/- is not shown, does it have a normal tumor weight comparable to wild type? Also, the graphical representation in figure 2c had two panels of NF-YA-/- .Please make sure it is correct.

We thank the reviewer for the comment. The results are shown in the graph in Figure 2c (photo: data not shown). The average tumor volume for *NFYA*^{+/+}, *NFYA*^{+/-}, and *NFYA*^{-/-} is 2250.3 mm³, 806.6 mm³, and 93.3 mm³, respectively, clearly placing the *NFYA*^{+/-} tumor volume in the middle of *NFYA*^{+/+} and *NFYA*^{-/-}. Also, since the graph in Figure 2c was complicated, we combined *NFYA*^{-/-} into one and changed the graph from tumor weight to tumor volume, as pointed out in comment 6.

Reviewer Figure 5 (Fig. 2c)

a Representative image of tumours formed in mammary fat pads of NOD/SCID mice 50 days after implanted with *NFYA*^{+/+} and *NFYA*^{-/-} SUM159 cells. Dot plots show the volume of tumours from each experimental group (n=10; *NFYA*^{+/+} and *NFYA*^{-/-}, n=7; *NFYA*^{+/-}).

6. In figure 2f, NFYAv2 transfection also increased tumor size compared to mock, the authors have not provided suitable explanation for this observation. Also, the graphical representation of tumor weight of NFYAv1 showed wide difference (figure 2f), what could be the reason. It is recommended to plot tumor volume rather than tumor weight.

We thank the reviewer for this constructive comment. We have re-plotted by tumour volume rather than tumour weight, as recommended. The results show no statistical difference between Mock and NFYAv2, although there is still a possibility that NFYAv2 may increase tumour size by partially complementing the function of NFYAv1. Our results showed a wide difference among NFYAv1 samples, possibly due to the combined effects such as mouse condition and injection area in the mammary fat pad. However, we used many mice to minimize these causes' influence on our conclusion. Since statistical differences were observed, we do not believe that these influences affect our conclusion.

Reviewer Figure 6 (Fig. 2f)

a Representative image of tumours formed in mammary fat pads of NOD/SCID mice 50 days after implanted with *NFYA*^{-/-} SUM159 cells overexpressed each variant of NFYA. Dot plots show the volume of tumours from each experimental group (n=10; Mock, n=11; NFYAv1, n=13; NFYAv2).

7. In supplementary figure 3c, we find that FASN inhibitor cerulenin alone is sufficient to inhibit sphere forming ability irrespective of the presence or absence of NFYA. The authors claim that NFYA is responsible for turning on lipogenic genes like FASN and others. This concept is to be clarified.

We have shown that NFYA promotes lipogenesis by inducing transcription of lipogenic enzymes, ACACA and FASN. Cerulenin irreversible inhibits FASN by covalently binding to the active site of FASN, thereby stopping the elongation reaction by malonyl-CoA condensation (Johansson *et al.*, 2008). Therefore, the action of Cerulenin is a post-response to FASN transcriptional regulation by NFYA and is independent of the presence or absence of NFYA.

8. In results 3, the authors elaborate the role of Palmitic acid in increasing OCR in Wild type and KO cells. However, the lines 203-205 is not convincing, please make the statements clear for the readers.

We apologize for the incomprehensible text. We have rewritten the manuscript for clarity.

In line 208-224:

“We also investigated these phenotypic properties by measuring the oxygen consumption rate (OCR) since *de novo* fatty acid synthesis is traded off with fatty acid oxidation (FAO) (Figure 3d, e). It is because malonyl-CoA inhibits carnitine palmitoyl transferase (CPT), which is the rate-limiting enzyme of FAO. Long-chain fatty acids, the major substrates fueling TCA cycle via degradation by FAO, increase the OCR. Therefore, we evaluated the fatty acid consumption by measuring the increase in OCR with the addition of BSA-conjugated palmitate. Under normal conditions, wild-type NFYA cells respond to palmitate supplementation by incrementing OCR, while NFYA-deficient cells showed less response to supplementation. These results indicated that FAO was limited by fatty acid synthesis and that NFYA-deficient cells used glucose as the carbon source in the TCA cycle under normal conditions. A more significant difference was observed in maximal respiration under high-energy demand conditions with uncoupling agent FCCP treatment. Under maximal respiration conditions, reserved respiration capacity was not changed between NFYA wild-type and deficient cells, indicating that FAO activity is retained in NFYA-deficient cells.”

9. In lines 215-216, the authors claim that NFYA deficient cells are more sensitive to glucose deprivation than Wild type cells. The authors cannot make such a direct correlation with lipid metabolism from this qualitative experiment alone. It is hence recommended to check the levels of lipogenic genes upon glucose starvation in NFYA mutant cells to draw such a conclusion.

We thank the reviewer for the critical comment. We agree with the reviewer that our results alone cannot show a direct correlation with glucose metabolism and that this is an overstatement. Since a direct correlation with glucose metabolism is beyond the scope of this paper, we have removed the results of Supplementary Fig. 3g from this manuscript. We intend to clarify them through future studies.

10. In lines 235-237, the immunofluorescence explanation on FASN in NFYA -/- supplemented with supplementary Figure 4a is of very low resolution. Replace the image to substantiate the theory.

We agree with the reviewer's comment about the low resolution of the images and have replaced them with higher-resolution images.

Reviewer Figure 7 (Supplementary Fig. 6a, b)

a, b $NFYA^{+/+}$ and $NFYA^{-/-}$ SUM159 cells were immunostained with anti-FASN (**a**, green) or anti-ACACA (**b**, green) antibodies and counterstained with DAPI for DNA (blue). A bar graph shows the relative fluorescence intensity of FASN or ACACA staining to DAPI.

11. In line 241-243, the authors claim that Etomoxir (ETO) increases CPT1A expression. This in turn is highly contradictory as ETO has been shown to be an inhibitor of CPT1 in line 218. Similarly cerulenin a reported inhibitor of FASN was used for the experiment, but the western blot images provided in Supplementary figure 4d don't show a significant down-regulation of FASN protein expression. Provide suitable explanation.

We thank the reviewer for the insightful comment. Etomoxir inhibits acyl-CoA transport from the cytoplasm to mitochondria by irreversibly inhibiting CPT1. Our results showed an unexpected increase in CPT1A expression upon Etomoxir treatment. The increased expression may reflect a compensatory response to inhibiting fatty acid oxidation. Similar increases in CPT1A expression have been reported in previous papers, and this compensatory increase in CPT1A protein is inactive (Schlaepfer *et al.*, 2014; Schlaepfer *et al.*, 2015; Petővári *et al.*, 2018). Cerulenin irreversibly inhibits FASN by covalently binding to the active site cysteine C1305 in the ketoacyl-synthase domain (KS) of FASN, thereby stopping the elongation reaction by malonyl-CoA condensation (Johansson *et al.*, 2008). Therefore, inhibition of FASN by Cerulenin is inhibition of enzyme activity and does not cause a decrease in protein expression. We have added the following text.

In line 253-265:

“As the inhibition of CPT1A by etomoxir completely abolished the effects of lipid addition on proliferation enhancement (Fig. 3f, g), we analyzed the expression of FAO-related factors to confirm that FAO was suppressed by etomoxir treatment. In both wild-type and NFYA-deficient cells, etomoxir treatment unexpectedly increased CPT1A expression (Supplementary Fig.6e). However, no difference in the expression pattern was observed with etomoxir treatment in the presence or absence of NFYA. It has been reported that increased CPT1A expression with etomoxir treatment may reflect a compensatory response to FAO suppression, but it is inactive³⁷⁻³⁹. In contrast, the FASN inhibitor cerulenin significantly reduced CPT1A expression only in NFYA wild-type cells, suggesting that NFYA regulates breast cancer malignant behavior by controlling *de novo* lipogenesis by regulating ACACA and FASN expression.”

12. Provide an appropriate reasoning for the expression of CD36 expression (supplementary figure 4f) in lines 263-267.

We thank the reviewer for the insightful comment. Similar to our results, it has been previously reported that inhibition or knockout of FASN increases CD36 expression (Feng et al., Cell Rep 2019, Drury et al., Front Oncol 2020). This increase in CD36 may be a compensatory response to fatty acid synthesis disruption. However, the current understanding of the regulation of CD36 expression is limited, and it is unclear how CD36 expression is regulated in cancer.

We showed by ChIP assay that the binding of NFYA to the promoter of FASN is completely abolished by lipid addition (Figure 4e). Conversely, we showed that the expression of CD36 is upregulated by lipid addition (Supplementary Fig. 7b). These results suggest that the expression of CD36 and FASN is determined by the amount of fatty acids present outside the cell, which may play a role in the regulation of these gene expressions.

However, since CD36 is highly expressed in NFYA-deficient cells, it is clear that NFYA does not directly regulate CD36 transcription, and a detailed analysis of the regulation of CD36 expression is beyond the scope of this paper and will be addressed in future studies.

Reviewer Figure 8 (Supplementary Fig. 7a, b)

a, b Western blot analysis of fatty acid translocase (CD36) in *NFYA*^{+/+} and *NFYA*^{-/-} SUM159 cells without (a) or with (b) lipid addition.

13. The authors have not elaborated the purpose of using various cell lines and compounds in detail. It makes the idea complicated for readers. So it is recommended to improve the language of the manuscript. There are a few grammatical errors which need to be rectified. References need to be made uniform.

We thank the reviewer for the comment. We have improved and revised the manuscript.

References

Zong, F. Y. *et al.*, The RNA-binding QKI suppresses cancer-associated aberrant splicing. *PLoS Genet.* **10**, e1005289; 10.1371/journal.pgen.1004289 (2014).

Nieto, M. A., Huang, R. Y., Jackson, R. A. & Thiery, J. P. EMT: 2016. *Cell* **166**, 21-45 (2016).

Yang, Y. *et al.*, Determination of a Comprehensive Alternative Splicing Regulatory Network and Combinatorial Regulation by Key Factors during the Epithelial-to-Mesenchymal Transition. *Mol. Cell Biol.* **36**, 1704-1719 (2016).

Johansson, P. *et al.* Inhibition of the fungal fatty acid synthase type I multienzyme complex. *Proc. Natl. Acad. Sci. USA* **105**, 12803-12808 (2008).

Schlaepfer, I. R. *et al.* Lipid catabolism via CPT1 as a therapeutic target for prostate cancer. *Mol. Cancer Ther.* **13**, 2361-2371 (2014).

Schlaepfer, I. R. *et al.* Inhibition of lipid oxidation increases glucose metabolism and enhances 2-Deoxy-2-[18F] Fluoro-D-Glucose uptake in prostate cancer mouse xenografts. *Mol. Imaging Biol.* **17**, 529-538 (2015).

Petővári, G. *et al.* Targeting cellular metabolism using rapamycin and/or doxycycline enhances anti-tumour effects in human glioma cells. *Cancer Cell Int.* **18**, 211; 10.1186/s12935-018-0710-0 (2018).

Feng W. W. *et al.* CD36-mediated metabolic rewiring of breast cancer cells promotes resistance to HER2-targeted therapies. *Cell Rep.* **29**, 3405-3420 (2019).

Drury J. *et al.* Inhibition of fatty acid synthase upregulates expression of CD36 to sustain proliferation of colorectal cancer cells. *Cell Rep.* **10**, 1185; 10.3389/fonc2020.01885 (2020).

REVIEWERS' COMMENTS:

Reviewer #1 (Remarks to the Author):

The authors have addressed my concerns and, for my part, the manuscript is now ready for publication.

I've also looked at the final manuscript and the comments from Reviewer 3. I would say that the rebuttal is appropriate and good: the authors have included additional and better experiments (in vivo tumour burden, new western blot analysis, higher resolution images of cells etc), but more importantly, they have contextualised some of their claims better and modified the text where appropriate. I'd be happy for the paper to be published